# Selective activation of FZD7 promotes mesendodermal differentiation of human pluripotent stem cells

Diana Gumber[1†], Myan Do[1†], Neya Suresh Kumar[1], Pooja R Sonavane[1], Christina C N Wu[2], Luisjesus S Cruz[3], Stephanie Grainger[3], Dennis Carson[2], Terry Gaasterland[4], Karl Willert[1]*

[1]Department of Cellular & Molecular Medicine, University of California San Diego, San Diego, United States; [2]Department of Medicine, University of California San Diego, San Diego, United States; [3]Department of Biology, San Diego State University, San Diego, United States; [4]University of California San Diego and Scripps Institution of Oceanography, Scripps Genome Center, La Jolla, United States

**Abstract** WNT proteins are secreted symmetry breaking signals that interact with cell surface receptors of the FZD family to regulate a multitude of developmental processes. Studying selectivity between WNTs and FZDs has been hampered by the paucity of purified WNT proteins and by their apparent non-selective interactions with the FZD receptors. Here, we describe an engineered protein, called F7L6, comprised of antibody-derived single-chain variable fragments, that selectively binds to human FZD7 and the co-receptor LRP6. F7L6 potently activates WNT/β-catenin signaling in a manner similar to Wnt3a. In contrast to Wnt3a, F7L6 engages only FZD7 and none of the other FZD proteins. Treatment of human pluripotent stem (hPS) cells with F7L6 initiates transcriptional programs similar to those observed during primitive streak formation and subsequent gastrulation in the mammalian embryo. This demonstrates that selective engagement and activation of FZD7 signaling is sufficient to promote mesendodermal differentiation of hPS cells.

**\*For correspondence:**
kwillert@ucsd.edu

†These authors contributed equally to this work

**Competing interests:** The authors declare that no competing interests exist.

## Introduction

WNTs are highly conserved, lipid-modified secreted proteins with a broad range of activities throughout development and during adult tissue homeostasis (reviewed in *Clevers et al., 2014*). Deregulated WNT activity is associated with many pathologies, including degenerative and age-related diseases and cancer (reviewed in *Nusse and Clevers, 2017*). With the human genome encoding 19 WNTs and an equally large number of WNT receptors (Frizzled [FZD]1–10, LRP5 and 6, ROR1 and 2, RYK, PTK7 and more; reviewed in *Niehrs, 2012*), relatively little is known about signaling specificities between WNT ligands and their receptors. In experimental settings, activation of WNT signaling with either purified recombinant WNT proteins, such as Wnt3a, or with small molecule agonists, such as GSK3 inhibitors, frequently elicits the desired downstream transcriptional effect. Such observations have led to the prevailing and oversimplified view that specific WNT-receptor interactions are not as critical as the ensuing downstream signaling event. Because of the lack of selectivity between WNTs and their receptors in vitro and the paucity of purified and biologically active WNT proteins, it has not been possible to address whether engagement of a single WNT receptor is sufficient to elicit the complex biological processes affected by WNT.

Human pluripotent stem (hPS) cells provide a powerful in vitro system to study early processes of human development. The role of WNT signaling in the regulation of pluripotency is dependent on

both the developmental stage of the cells and on the level of signaling. Experiments in mouse embryonic stem cells, which reside in the naive state (reviewed in *Smith, 2017*), have indicated a role for WNT/β-catenin signaling in self renewal and maintenance of pluripotency (*Hao et al., 2006*; *Ogawa et al., 2006*; *Sato et al., 2004*; *ten Berge et al., 2011*). In contrast, in hPS cells, which reside in the primed state and resemble epiblast-derived stem cells (*Tesar et al., 2007*), WNT/β-catenin signaling drives mesendodermal lineage specification (*Davidson et al., 2012*; *Sumi et al., 2008*; *Tsakiridis et al., 2014*), and an analysis of transcriptome-wide gene expression changes over a time-series treatment of hPS cells with Wnt3a reveals a transcriptional response reminiscent of primitive streak formation and ensuing gastrulation (*Huggins et al., 2017*). This is consistent with observations that short-term activation of Wnt/β-catenin signaling, via addition of Wnt3a, promotes differentiation toward definitive endoderm (DE) (*D'Amour et al., 2006*). Furthermore, a low level of endogenous WNT activity is detectable in undifferentiated hPS cells (*Blauwkamp et al., 2012*; *Jiang et al., 2013*) and is required for reprogramming of human fibroblasts to an induced pluripotent stem (iPS) cell state (*Ross et al., 2014*).

The WNT receptor FZD7 plays a prominent role in the regulation of pluripotency of hPS cells (*Fernandez et al., 2014*; *Melchior et al., 2008*), and its downregulation accompanies differentiation and exit from the pluripotent stem cell state. However, since hPS cells express several WNT receptors, including FZD2, 3, and 5 (*Fernandez et al., 2014*; *Huggins et al., 2017*), it is unclear whether Wnt3a promotes mesendodermal differentiation through FZD7 or whether other FZDs are required.

Here, we describe the development and characterization of a highly specific antibody and derivative single-chain variable fragment (scFv) to FZD7. Fusing this FZD7-specific scFv to an LRP6-directed scFv creates a bispecific binder, called F7L6, with potent WNT signaling activities that require FZD7 expression. Treatment of hPS cells with F7L6 elicits a transcriptional response similar to that observed for Wnt3a treatment, establishing that signaling through FZD7 is sufficient to promote mesendodermal differentiation.

## Results

### Design, specificity, and expression of F7L6

WNT/β-catenin signaling is initiated by WNT binding to and heterodimerizing the cell surface receptors FZD and either LRP5 or LRP6. To study the role of one specific FZD protein, FZD7, which is highly expressed in hPS cells (*Fernandez et al., 2014*; *Melchior et al., 2008*), we developed a bispecific binder to human FZD7 and LRP6, called F7L6. We generated a single-chain variable fragment (scFv) to FZD7 by repurposing the variable region of our FZD7-targeting antibody (F7-Ab), a chimeric human/mouse IgG1. This antibody was derived from a bacterially produced antigen binding fragment (Fab) described in a previous study (*Fernandez et al., 2014*). In parallel, we engineered an scFv to the third β-propeller domain of LRP6 based on previously published LRP6 antibodies (*Ettenberg et al., 2010*). We then engineered F7L6 by conjugating the FZD7-scFv and LRP6-scFv to a human IgG1 Fc to create the FZD7-specific Wnt mimetic F7L6 (complete amino acid sequence is provided in *Figure 1—figure supplement 1*). Additionally, we generated a FZD7-scFv–Fc (F7) and an LRP6-scFv-Fc (L6) (*Figure 1A*). We confirmed secretion of F7-Ab, F7, L6, and F7L6 from CHO cells by SDS-PAGE and Coomassie Blue staining and immunoblotting (*Figure 1B*). Using a dot blot with lysates of HEK293T cells carrying loss-of-function mutations in FZD1, 2, and 7 (F127-KO) (*Voloshanenko et al., 2017*) or in LRP6 (LRP6-KO) (*Grainger et al., 2019*) confirmed binding specificity of each of these molecules. FZD7-specific binders (F7 and F7L6) only reacted with lysates upon FZD7 overexpression (note that endogenous FZD7 expression in LRP6 KO is below the level of detection), whereas LRP6-specific binders (L6 and F7L6) only reacted with lysates of cells expressing LRP6 (*Figure 1C*).

To further characterize the specificity of F7-Ab, we mapped its binding site to the 'neck' region between the cysteine-rich domain (CRD) and the first transmembrane domain of FZD7 (*Figure 1D*). Despite the high degree of homology between human and mouse Fzd7 (hFZD7 and mFzd7, respectively), F7-Ab only reacts with hFZD7 (*Figure 1E*). Protein alignment (*Figure 1D*) indicated that this neck region was the only extracellular portion harboring differences between hFZD7 and mFzd7 (disregarding the amino-terminal signal sequence). A single amino acid change at position 188 from leucine to proline (P188) renders hFZD7 non-reactive to F7-Ab, whereas the corresponding amino acid

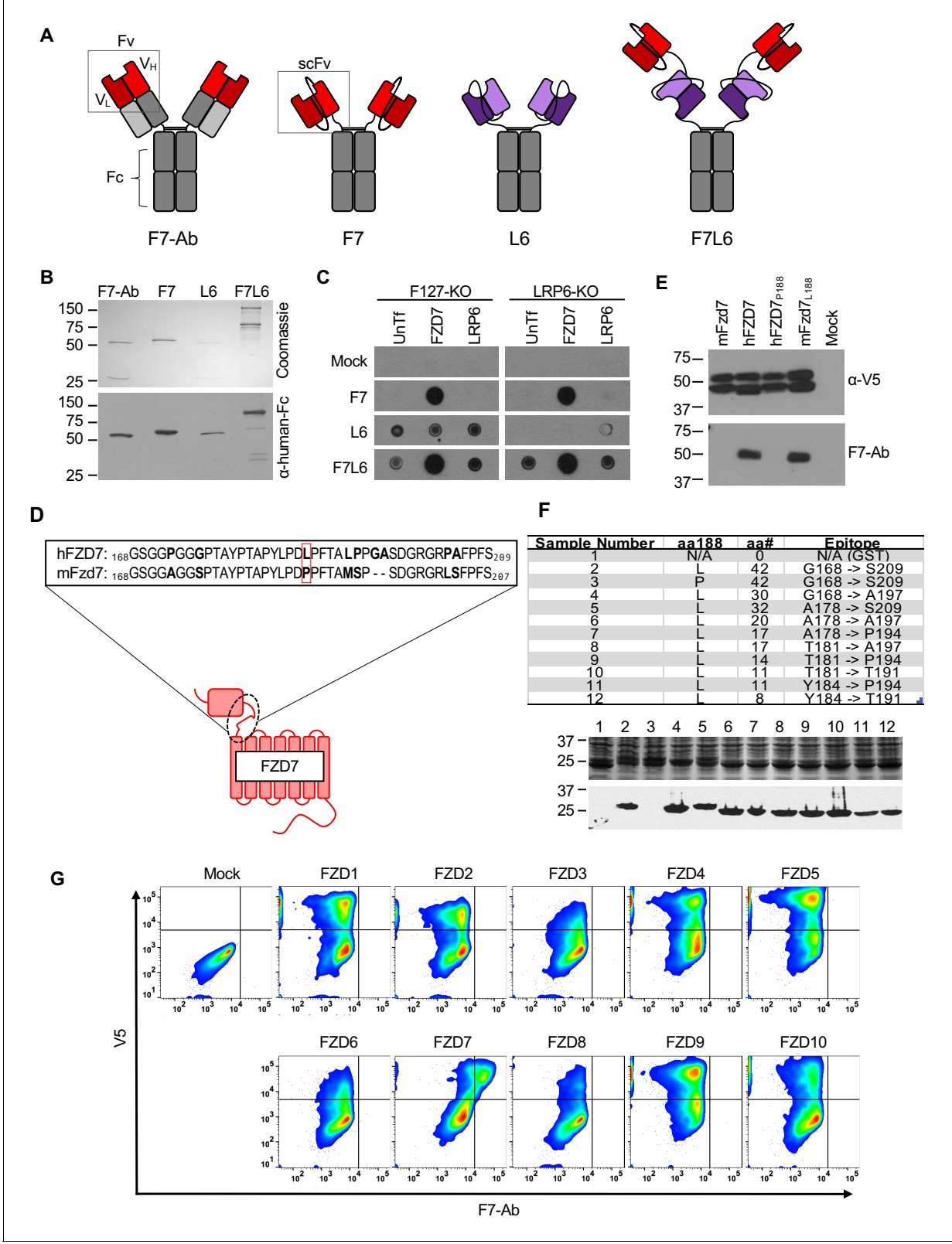

**Figure 1.** Design, specificity, and expression of F7L6. (**A**) Schematic of FZD7- and LRP6-specific binders. Red blocks depict the variable light ($V_L$) and heavy ($V_H$) antibody domains that recognize FZD7 and were fused to form the single-chain variable fragment (scFv) in F7 and F7L6. Purple blocks depict the scFv that recognizes LRP6 and is used in L6 and F7L6. (**B**) Expression of FZD7- and LRP-specific binders. Transgenes encoding F7-Ab, F7, L6, and F7L6 were stably transduced in CHO cells. Recombinant proteins were harvested and purified from conditioned media and detected by Coomassie

*Figure 1 continued on next page*

*Figure 1 continued*

Blue staining (upper) and anti-human-Fc immunoblot (lower). (**C**) Binding specificity of F7, L6, and F7L6. HEK293T carrying mutations in FZD1, 2, and 7 (F127-KO) or in LRP6 (LRP6-KO) were transfected with FZD7 and LRP6, respectively, and whole cell lysates were probed in a dot blot format with conditioned media containing F7, L6, or F7L6. As a negative control (Mock), blots were incubated with CM from untransfected CHO cells. (**D**) Schematic of FZD7 and amino acid alignment of the extracellular 'neck' region of hFZD7 and mFzd7. The dashed oval indicates the neck region. The red box in the amino acid sequences indicates amino acid position 188. (**E**) HEK293T cells were transiently transfected with the indicated V5-tagged transgenes, and cell lysates were probed with either F7-Ab or V5 antibody (α-V5). Mock = untransfected cells. (**F**) Mapping F7-Ab epitope to an eight amino acid sequence. Bacterial lysates containing fusion proteins between GST and the FZD7 peptide sequences indicated in the table were detected by Coomassie staining (top) or by immunoblotting with F7-Ab (bottom). Abbreviations in table: aa188, amino acid at position 188; aa#, number of amino acids in FZD7 peptide; L, leucine; P, proline. (**G**) F7-Ab is specific to human FZD7 and does not cross-react with the other nine FZDs (1-6, 8-10). F127-KO were transfected with expression constructs carrying the indicated human FZD cDNAs tagged with an intracellular V5 sequence. Non-permeabilized cells were stained with F7-Ab for cell-surface FZD expression, and then permeabilized and stained for V5 expression. All FZD receptors were expressed as revealed by anti-V5 antibody staining.

The online version of this article includes the following figure supplement(s) for figure 1:

**Figure supplement 1.** Complete amino acid sequence of His-tagged F7L6.
**Figure supplement 2.** Immunofluorescent detection of V5-tagged FZD1-10 demonstrates cell surface localization.

change in mFzd7 (L188) restores F7-Ab reactivity (*Figure 1E*). Using fusion proteins between gluta-thione S-transferase (GST) and the FZD7 neck region followed by sequential shortening of the neck region, we mapped the F7-Ab epitope to an eight amino acid stretch containing L188 (*Figure 1F*). Since FZD proteins are highly conserved and several available FZD antibodies react with multiple FZD proteins (for example, OMP-18R5/Vantictumab reacts with FZD1, 2, 5, 7, and 8), we confirmed that F7-Ab does not cross-react with the other nine human FZD receptors (FZD1-6, 8–10) (*Figure 1G*). To rule out the possibility that some FZDs fail to react with F7-Ab because they do not reach the cell surface, we performed confocal microscopy. This analysis confirmed that all FZD proteins were expressed on the cell surface (*Figure 1—figure supplement 2*). Taken together, F7-Ab, and hence F7 and F7L6, are specific to hFZD7 and do not cross-react with any of the other nine human FZD proteins or with mFzd7.

## F7L6 activates Wnt/β-catenin signaling

The bispecific binder, F7L6, activates WNT/β-catenin signaling by heterodimerizing FZD7 and LRP6 (*Figure 2A*). We confirmed F7L6 signaling activity using HEK293T cells stably transfected with the WNT reporter Super TOP-Flash (STF) (*Veeman et al., 2003*; *Figure 2B*). As expected, single binders to FZD7 (F7) or LRP6 (L6) did not activate signaling. As is the case with native WNT proteins, activity of F7L6 is potently augmented by addition of R-Spondin1 (RSPO1) (*Kim et al., 2008*; *Figure 2C*) with potency in the single-digit nanomolar range. Interestingly, addition of RSPO1 increased F7L6 activity approximately 10-fold, but only augmented the activities of Wnt3a or FLAg $F^{P+P}$-$L^{1+3}$, a previously published WNT mimetic (*Tao et al., 2019*), by twofold (*Figure 2—figure supplement 1*). A possible reason for this distinction is that F7L6 is selective for FZD7, whereas Wnt3a and FLAg $F^{P+P}$-$L^{1+3}$ are capable of interacting with multiple FZDs. Furthermore, both Wnt mimetics retain significant activity at sub-nanomolar concentrations, whereas Wnt3a's activity is undetectable at such concentrations.

In contrast to WNT, which binds to the CRD of FZD (*Janda et al., 2012*), F7L6 binds to the neck region between the CRD and first transmembrane domain, allowing us to assess the requirement of the CRD in signaling. Interestingly, FZD7 lacking the CRD (CRD-less FZD7) activates signaling when heterodimerized to LRP6 with F7L6 while Wnt3a does not (*Figure 2D*), indicating that the CRD is dispensable for signaling. Furthermore, appending the F7-Ab epitope of eight amino acids (see *Figure 1F*) onto another FZD, FZD2, is sufficient for F7L6 to activate WNT/β-catenin signaling through FZD2 (*Figure 2E*). These data establish that heterodimerization of FZD7 or FZD2 with LRP6 is sufficient for pathway activation.

Treatment of mouse L-cells with Wnt3a activates downstream signaling, as assayed by cyto-plasmic β-catenin stabilization (*Shibamoto et al., 1998*), whereas treatment with F7L6 does not (*Figure 2F*), consistent with the lack of reactivity of F7-Ab with mouse Fzd7 (see *Figure 1E*). However, overexpression of human FZD7 in these cells is sufficient to activate signaling by F7L6 (*Figure 2F*), thus confirming F7L6 signaling specificity through FZD7. The kinetics of β-catenin

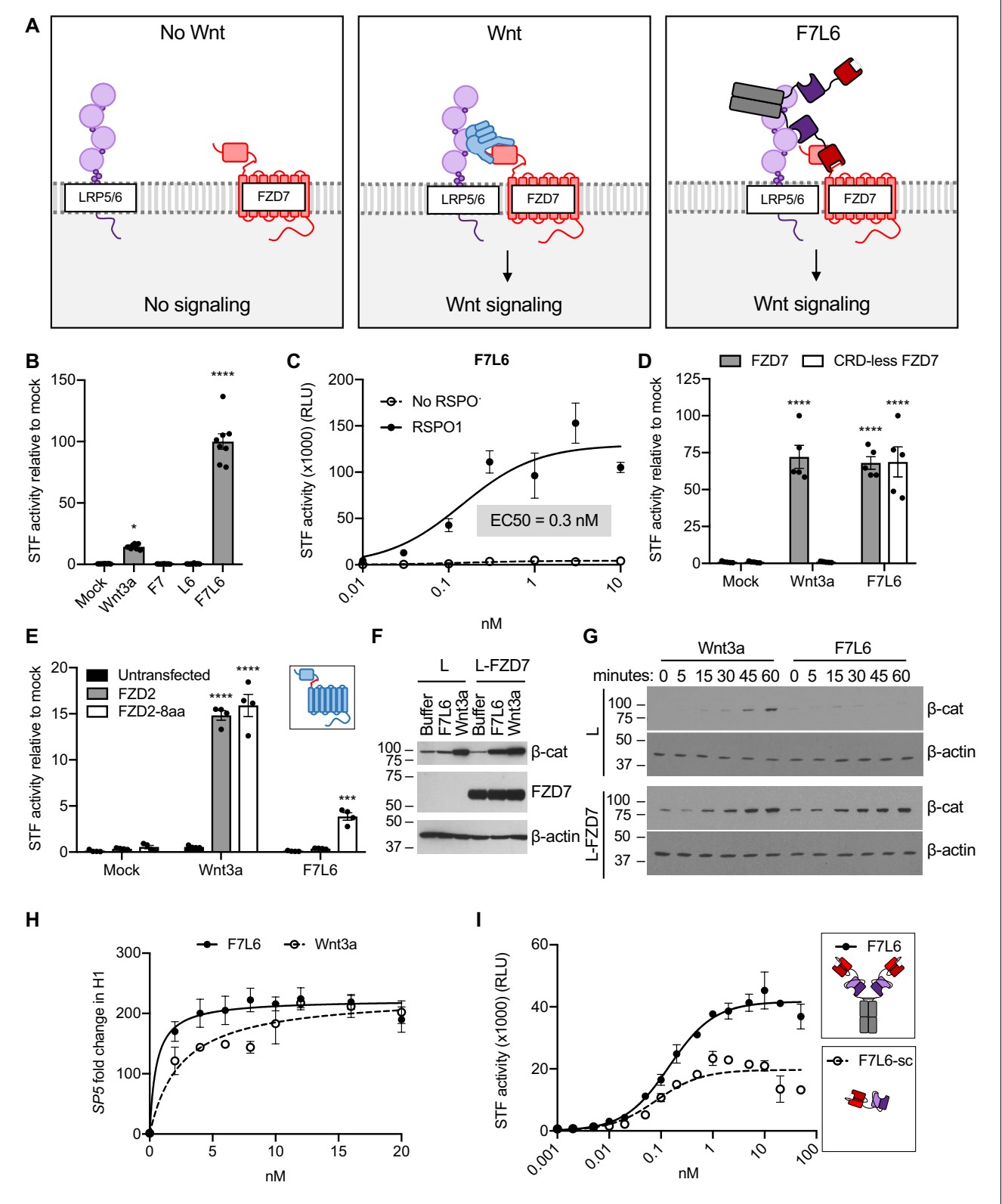

**Figure 2.** F7L6 activates Wnt/B-catenin signaling. (**A**) Schematic of the proposed mechanism of action of F7L6 through the heterodimerization of FZD7 and LRP6 at the cell surface. (**B**) F7L6 activation of the WNT signaling pathway was evaluated using a luciferase-based WNT reporter (Super TOP-Flash, STF) assay. HEK293T stably transduced with the WNT reporter Super TOP-Flash (STF) were treated with indicated conditioned media for 24 hr and then assayed for luciferase activity. (**C**) F7L6 signaling activity is augmented by RSPO1. HEK293T:STF cells were treated with the indicated concentrations of

*Figure 2 continued on next page*

*Figure 2 continued*

F7L6 in the presence or absence of RSPO1 (100 ng/mL) for 24 hr and then assayed for luciferase activity (RLU = relative light units). (D) F7L6 activates signaling independently of the WNT-binding cysteine-rich domain (CRD). F127-KO cells carrying the STF reporter were transfected with expression plasmids carrying wildtype FZD7 or CRD-less FZD7, treated with Wnt3a or F7L6 for 24 hr and then assayed for luciferase activity. Inset illustrates FZD7 lacking the CRD. (E) F7L6 activates FZD2 tagged with the eight-amino acid epitope of FZD7. F127-KO cells carrying the STF reporter were transfected with expression plasmids carrying wildtype FZD2 or FZD2-8aa, treated with Wnt3a or F7L6 for 24 hr and then assayed for luciferase activity. Inset illustrates FZD2 (blue) with the eight-amino acid FZD7 tag (red). (F) F7L6 leads to β-catenin stabilization in mouse L-cells expressing human FZD7. Untransfected (L) or FZD7-expressing (L-FZD7) L-cells were treated with 10 nM F7L6 or Wnt3a for 3 hr. Cell lysates were immunoblotted for β-catenin and FZD7. Blotting for β-actin served as a loading control. (G) F7L6 leads to β-catenin stabilization in a time-dependent manner. L and L-FZD7 cells were treated with 10 nM Wnt3a or F7L6 for the indicated times, and cell lysates were immunoblotted for β-catenin. Blotting for β-actin served as a loading control. (H) F7L6 activates *SP5* expression in hPS cells in a dose-dependent manner. H1/WA01 cells were treated with the indicated doses of F7L6 or Wnt3a for 24 hr. RNA was analyzed by qRT-PCR. Data represented as mean ± SEM for three technical replicates, with a nonlinear regression curve. All samples were normalized to the 0 nM (buffer) control. (I) Bivalent and tetravalent Wnt mimetics activate Wnt signaling. HEK293T:STF cells were treated with indicated concentrations of either F7L6 (tetravalent) or F7L6-sc (bivalent) in the presence of RSPO1 (100 ng/mL) for 24 hr and then assayed for luciferase activity (RLU = relative light units). For all statistical analyses: one-way ANOVA and Tukey's multiple comparisons test: ****p≤0.0001, *p≤0.05.

The online version of this article includes the following source data and figure supplement(s) for figure 2:

**Source data 1.** Raw data for STF and RT-qPCR assays shown in *Figure 2*.
**Figure supplement 1.** Dose response curves for a previously published Wnt mimetic, FLAg$^{P+P}$-L6$^{1+3}$ (described in *Tao et al., 2019*) and Wnt3a.
**Figure supplement 2.** F7L6 activates Wnt/β-catenin signaling.
**Figure supplement 3.** Size exclusion chromatography (SEC) of F7L6 and F7L6-sc.

---

stabilization in L-cells is similar for Wnt3a and F7L6, being first detectable between 15 and 30 min after stimulation (*Figure 2G*). Lastly, using reverse transcription quantitative PCR (RT-qPCR), we demonstrated that F7L6, like Wnt3a, activates expression of the WNT target gene, *SP5*, in hPS cells, which express FZD7 (*Fernandez et al., 2014*; *Melchior et al., 2008*), in a dose-dependent manner (*Figure 2H*, *Figure 2—figure supplement 2*). In sum, F7L6 potently induces WNT/β-catenin signaling through FZD7.

F7L6 is a bispecific bivalent (or tetravalent) molecule, potentially capable of simultaneously engaging two FZD7 and two LRP6 molecules, thus leading to receptor oligomerization, which has been proposed to promote intracellular signalosome formation (*DeBruine et al., 2017*). To address whether FZD7-LRP6 heterodimerization alone is sufficient for signaling, we deleted the Fc portion to generate a single-chain F7L6 (F7L6-sc) that is predicted to engage one FZD7 and one LRP6 receptor. F7L6-sc activated Wnt signaling, albeit to a lesser extent than the tetravalent F7L6 protein (*Figure 2I*). Interestingly, the maximal response in the STF assay for F7L6 was approximately twice that for F7L6-sc. Size exclusion chromatography confirmed that F7L6-sc is monomeric in solution and does not form larger oligomers (*Figure 2—figure supplement 3*). These data demonstrate that a 1:1 association of FZD7 and LRP6 is sufficient to activate downstream signaling.

## Differential effects of F7L6, Wnt3a, and CHIR on gene expression in hPS cells

Previous studies have shown that modulating WNT/β-catenin signaling is crucial for differentiation of hPS cells into mesendoderm and DE (*D'Amour et al., 2006*; *Huggins et al., 2017*; *Jiang et al., 2013*; *Martyn et al., 2018*; *Yoney et al., 2018*). Wnt3a and GSK3-inhibitors (e.g. CHIR99021, CHIR98014, BIO, Li) (*Klein and Melton, 1996*; *Ring et al., 2003*; *Sato et al., 2004*; *Stambolic et al., 1996*) are frequently used interchangeably as WNT pathway activators in these differentiation protocols (*Gertow et al., 2013*; *Huang et al., 2017*; *Kumar et al., 2015*; *Loh et al., 2014*; *Naujok et al., 2014*; *Teo et al., 2014*). Although Wnt3a is an established activator of WNT/β-catenin signaling, relatively little is known about the receptors that promote WNT-driven differentiation of hPS cells. We hypothesized that FZD7 mediates this process, because it transduces the Wnt3a signal (*Dijksterhuis et al., 2015*; *Voloshanenko et al., 2017*) and regulates the pluripotent state (*Fernandez et al., 2014*; *Melchior et al., 2008*). Here, we used F7L6 to selectively activate FZD7 signaling in hPS cells and analyzed global changes in gene expression by RNA sequencing (RNA-seq). We also examined the temporal kinetics of gene expression changes in response to a single, continuous dose of F7L6, Wnt3a, or CHIR98014 (CHIR) treatment.

Clustering of significantly differentially expressed genes (1814 genes, *Figure 3—figure supplement 1*, *Supplementary file 1A*) according to change in percent maximum reads per kilobase per million mapped reads (RPKM) revealed distinct waves and classes of gene expression, including activation of many known Wnt pathway genes (*Figure 3A*), activation of mesendodermal and endodermal genes, and downregulation of pluripotency- and ectoderm-associated genes (*Figure 3B*).

Gene expression changes in response to Wnt3a and F7L6 were overall highly similar, with robust activation of classic markers of mesendoderm (i.e. *GSC*, *MIXL1*, *SP5* and *T*) and DE (i.e. *LEFTY1*, *EOMES*, *CER1*, *FOXA2*, *CXCR4*, and *SOX17*) and downregulation of pluripotency markers, such as *POU3F1*, *FZD7*, and *PODXL* (TRA-1–81). However, there were some notable differences between Wnt3a and F7L6 treatments. For example, induction of established WNT target genes, such as *DKK1*, *AXIN2*, *WLS*, *WNT3*, and *NKD1*, was more potent with Wnt3a than with F7L6. Also, induction of genes, such as *T*, *SP5*, *GSC*, and *FOXA2*, was slightly delayed for F7L6 relative to Wnt3a. These differences are potentially due to F7L6 solely engaging FZD7 and LRP6 to promote downstream signaling, while Wnt3a, in contrast, promotes signaling through multiple FZD-LRP5/6 heterodimers.

In contrast to Wnt3a and F7L6, CHIR treatment produced significantly different gene expression profiles, both in kinetics and amplitude (*Figure 3A,B*). For example, while all three treatments activated known WNT target genes (e.g. *AXIN2*, *DKK1*, *LEF1*, and *SP5*) and mesendodermal genes (e.g. *T*, *MIXL1*, *GATA4*, and *EOMES*), CHIR did so more robustly than Wnt3a or F7L6. Such differences are likely attributable to the fact that CHIR acts downstream of the receptor complexes and hence is not restricted by receptor availability. Furthermore, while all three treatments activated expression of Wnt pathway and mesendodermal genes, CHIR failed to activate expression of certain genes, in particular the DE markers *FOXA2* and *SOX 17* (and to a lesser extent *CER* and *GSC*), an effect we validated using RT-qPCR (*Figure 3—figure supplement 2*). Interestingly, expression of other DE markers, such as *CXCR4*, which encodes a cell surface receptor commonly used to enrich DE populations using flow cytometry (*D'Amour et al., 2005*; *McGrath et al., 1999*), was activated by CHIR (*Figure 3B*). These results indicate that sustained activation of WNT/β-catenin signaling by GSK3 inhibition with CHIR produces significant differences compared to selective pathway activation with either Wnt3a or F7L6.

F7L6, Wnt3, and CHIR treatment significantly altered expression of 525, 708, and 1646 genes, respectively (*Figure 4*, *Supplementary file 1B–D*). A majority of genes (428 of 805 genes) (*Supplementary file 1E and F*) in the F7L6 and Wnt3a lists overlapped, whereas expression changes of a large number of genes was unique to CHIR (1009 genes) (*Supplementary file 1G*), highlighting the substantial differences when the pathway is activated at the level of receptor rather than at the level of GSK3 inhibition. Using Gene Set Enrichment Analysis (GSEA) (*Mootha et al., 2003*; *Subramanian et al., 2005*) of the 391 genes (*Supplementary file 1H*) that changed in response to all three treatments indicated enrichment of diverse developmental processes, including Animal Organ Morphogenesis (Gene Ontology [GO]:0009887), Anatomical Structure Formation Involved In Morphogenesis (GO:0048646), Tube Development (GO:0035295), Circulatory System Development (GO:0072359), and Neurogenesis (GO:0022008) (*Figure 4*, *Supplementary file 2A*).

Importantly, GSEA of all genes altered by CHIR indicated enrichment in Neurogenesis (GO:0022008) and Neuron Differentiation (GO:0030182), both categories that were not among the top five GO terms for F7L6 and Wnt3a (*Figure 4*, *Supplementary file 2B–D*), indicating that CHIR treatment additionally promotes differentiation into ectodermal lineages. The 1009 differentially expressed genes unique to CHIR were additionally enriched in processes such as Biological Adhesion (GO:0022610), Locomotion (GO:0040011), Cell Junction (GO:0030054), and Positive Regulation Of Multicellular Organismal Process (GO:0051240) (*Supplementary file 2E*). In summary, our RNA-seq analyses confirmed that activation of FZD7 signaling by F7L6 was sufficient to modulate WNT target genes and induce expression of mesendoderm and DE differentiation in hPS cells. Overall, F7L6 and Wnt3a treatments induced similar (although not identical) patterns and temporal kinetics of gene expression, while CHIR caused much wider changes in global gene expression.

## Early WNT target gene activation by F7L6 and Wnt3a in hPS cells

As revealed by our RNA-seq analysis, selective FZD7 pathway activation elicits a complex program of gene expression. Recent studies have provided evidence that certain Wnts recruit additional receptors to the Fzd-Lrp5/6 complex to increase signaling specificity. For example, the cell surface proteins Reck and Gpr124 promote Wnt7 signaling through Fzd (*Cho et al., 2017*; *Eubelen et al.,*

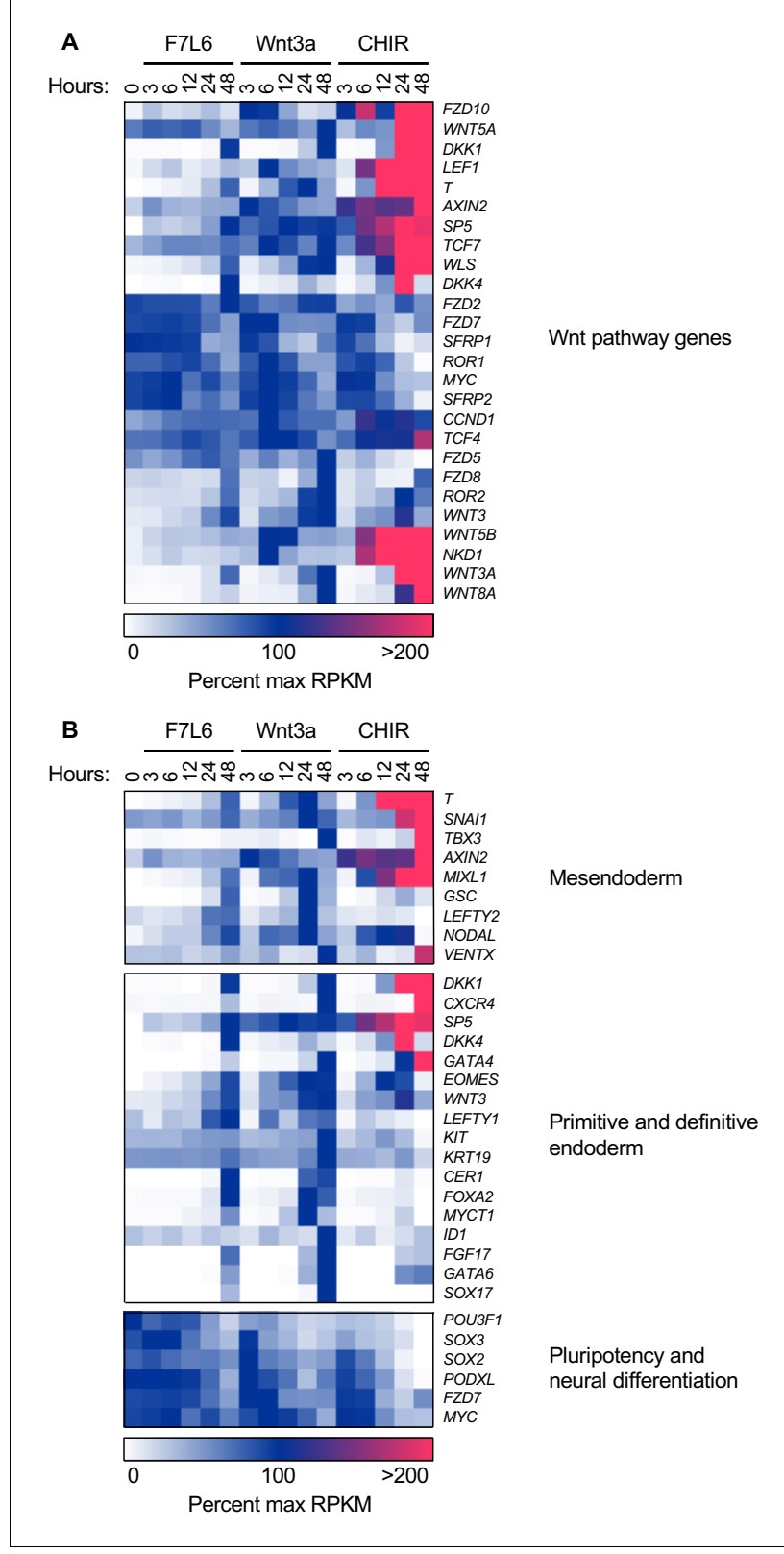

**Figure 3.** F7L6, Wnt3a, and CHIR differentially alter the transcriptome of human pluripotent stem (hPS) cells. hPS cells (H1/WA01) were treated with 5 nM F7L6 or Wnt3a, or 250 nM CHIR98014 (CHIR) for the indicated hours. RNA was isolated and analyzed by RNA-seq. Significant differential gene expression was defined as a 1.75-fold increase or decrease in RPKM compared to the 0 hr (buffer) control. Expression is represented as percent maximum RPKM

*Figure 3 continued on next page*

*Figure 3 continued*

(0, white; 100, blue;≥100, pink). RPKM for each gene was normalized to the maximum RPKM across F7L6 and Wnt3a treatment groups. Supplementary Data 1 provides complete gene list. (**A**) Heat map of Wnt target genes changed in response to F7L6, Wnt3a, or CHIR. (**B**) Heat map of changed genes associated with mesendoderm and primitive/definitive endoderm differentiation, and pluripotency and neural differentiation. F7L6 promotes mesendodermal differentiation, similarly to Wnt3a.

The online version of this article includes the following source data and figure supplement(s) for figure 3:

**Figure supplement 1.** F7L6, Wnt3a, and CHIR differentially alter the transcriptome of human pluripotent stem (hPS) cells.

**Figure supplement 2.** CHIR does not activate select definitive endoderm markers in undirected differentiation of human pluripotent stem cells.

**Figure supplement 2—source data 1.** Raw data for RT-qPCR assays shown in *Figure 3—figure supplement 2*.

---

*2018*). Furthermore, Egfr promotes a specific interaction of Wnt9a with Fzd9 and Lrp5/6 (*Grainger et al., 2019*). Likewise, Wnt3a may recruit additional, and currently unknown, co-receptors to activate signaling in hPS cells. In contrast, owing to its design, our Wnt mimetic F7L6 only recruits FZD7 and LRP6. To confirm that Wnt3a and F7L6 promote similar downstream signaling events, we more closely examined early WNT target gene activation and identified groups of genes with maximal activation at 3 hr and at 6 hr upon Wnt3a treatment (*Figure 5A and B*, *Supplementary file 3A*

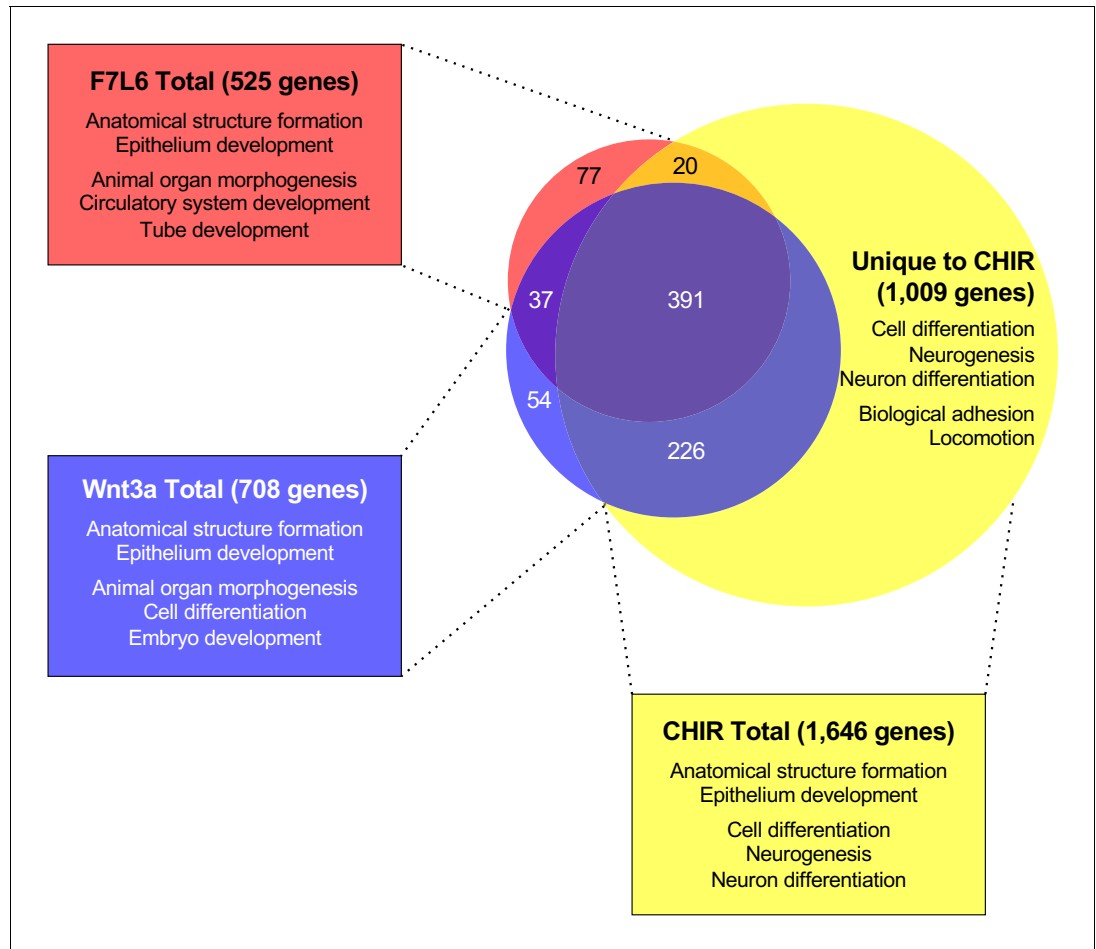

**Figure 4.** Gene Set Enrichment Analysis (GSEA) analyses of transcriptome changes induced by F7L6, Wnt3a, and CHIR. Venn diagram and GSEA analyses of genes differentially expressed in response to F7L6, Wnt3a, and CHIR98014 treatments in hPS cells (H1/WA01). The top five GSEA gene set hits for each treatment group are listed by commonality among the groups and alphabetical order. *Supplementary file 1* provides lists of gene names and *Supplementary file 2* provides lists of gene set names.

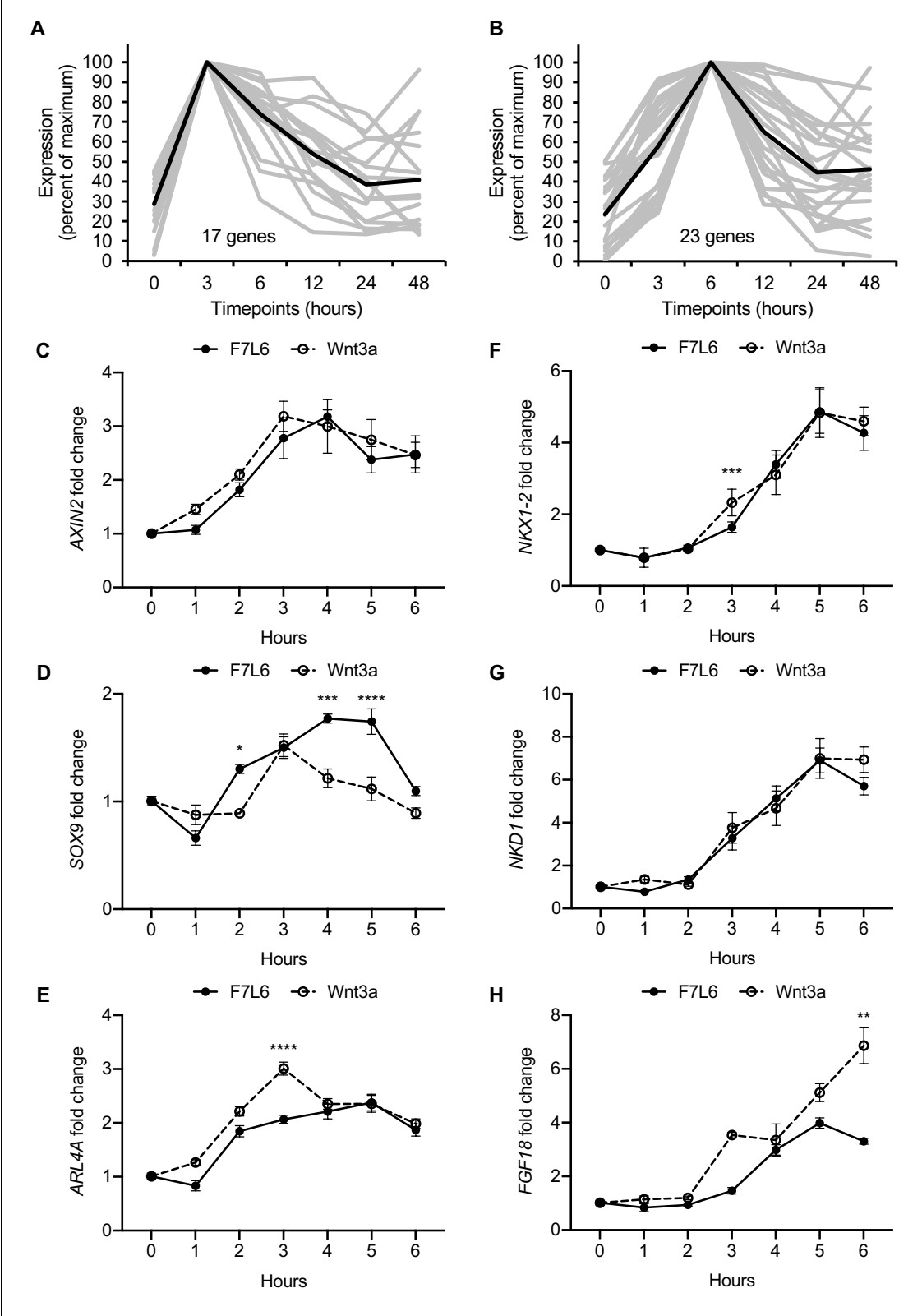

**Figure 5.** Early Wnt target genes activated by F7L6 and Wnt3a in hPS cells. RNA-seq profiles of genes significantly activated at 3 hr (**A**) and 6 hr (**B**) in response to Wnt3a (5 nM) in H1/WA01 cells. Lists of genes are provided in *Supplementary file 3 A and B*. Validation of three target genes (*AXIN2*, *SOX9*, and *ARL4A*) maximally activated at 3 hr (**C–E**) and of three genes (*NKX1-2*, *NKD1*, and *FGF18*) maximally activated at 6 hr (**F–G**). H1 cells were treated with F7L6 or Wnt3a (each at 10 nM) for the indicated time and total RNA was analyzed by RT-qPCR. Gene expression was normalized to the

*Figure 5 continued on next page*

*Figure 5 continued*

expression of *RPL13A*. Data represented as mean ± SEM for two independent experiments, three technical replicates each. All samples were normalized to the 0 hr (buffer) control. For statistical analyses: one-way ANOVA and Tukey's multiple comparisons test for significance between F7L6 and Wnt3a treatments at each time point: ****$p \leq 0.0001$, ***$p \leq 0.001$, **$p \leq 0.01$, *$p \leq 0.05$.

The online version of this article includes the following source data for figure 5:

**Source data 1.** Raw data for RT-qPCR assays shown in *Figure 5*.

*and B*). We used RT-qPCR to validate changes in expression of several genes within these clusters in response to Wnt3a and F7L6 over a 6-hr time course in hPS cells. Both Wnt3a and F7L6 activated these early target genes with similar kinetics, with expression of *AXIN2*, *SOX9*, and *ARL4A* at 3 hr (*Figure 5C–E*) and *NKX1-2*, *NKD1* and *FGF18* at 6 hr (*Figure 5F–H*). These findings support that heterodimerization of FZD7 and LRP6 by F7L6 is sufficient to elicit a transcriptional response in hPS cells similar to that elicited by Wnt3a and likely does not involve the recruitment of additional receptors.

## FZD7 activation promotes differentiation of hPS cells

Our RNA-Seq analysis indicated that F7L6, like Wnt3a, promotes mesendodermal differentiation. To monitor differentiation in real time, we employed human embryonic stem (hES) cells (H9/WA09) carrying reporter genes that mark meso- and endo-dermal differentiation: (1) for mesodermal differentiation, TBXT(T)-GFP, which harbors the gene encoding enhanced green fluorescent protein (eGFP) under control of the *TBXT* (T) promoter (*Kita-Matsuo et al., 2009*) and (2) for endodermal differentiation, SOX17-eGFP, which carries the eGFP gene in the *SOX17* locus (*Wang et al., 2011*). We treated these reporter cell lines with a single, continuous dose of F7L6, Wnt3a or CHIR and monitored GFP expression for 5 days (*Figure 6A*). F7L6 potently activated expression of both reporters (*Figure 6B,C*, *Figure 6—videos 1* and *2*), with T-GFP expression first detectable at 24 hr (*Figure 6B,D*) and SOX17-eGFP at 36 hr (*Figure 6C,E*). Wnt3a likewise activated expression of both transgenes with similar temporal kinetics, however, their induction was significantly lower than with F7L6 (*Figure 6B–E*). This difference in potency between F7L6 and Wnt3a is likely due to the fact that Wnt3a is more unstable than F7L6 in these culture conditions. Treatment with CHIR yielded reporter gene expression distinct from both F7L6 and Wnt3a, with an early peak in T-GFP expression that subsequently declines (*Figure 6B,D*) and no induction of the SOX17-eGFP reporter (*Figure 6C,E*). This is consistent with our RNA-Seq (*Figure 3B*) and RT-qPCR results (*Figure 3—figure supplement 2*), again highlighting the distinct effects on differentiation of GSK3 inhibition versus FZD7 activation with either Wnt3a or F7L6.

## FZD7 activation directs endodermal differentiation

Protocols to differentiate hPS cells specifically toward DE have been established and are widely used to generate mature endodermally derived cell populations, such as pancreas (*D'Amour et al., 2006*), liver (*Touboul et al., 2010*), and intestine (*Workman et al., 2017*). DE differentiation of hPS cells is induced by Activin/Nodal signaling, and Wnt3a addition on day 1 of differentiation increases the efficiency of mesendoderm specification and subsequent DE formation (*D'Amour et al., 2006*; *Figure 7A*). Using this protocol, we replaced Wnt3a treatment with F7L6 and monitored gene expression by RT-qPCR. As expected, expression of the pluripotency marker *OCT4* (*POU5F1*) declined over the 3 days of differentiation with a more potent effect by F7L6 compared to Wnt3a (*Figure 7B*). Concurrently, expression of endodermal markers, *SOX17*, *CXCR4* and *FOXA2*, increased upon treatment with F7L6 to a similar extent as Wnt3a (*Figure 7B*), indicating that activation of FZD7 alone is sufficient to promote DE formation.

## FZD7 activation prevents differentiation toward the hematopoietic lineage

Hematopoietic stem and progenitor cells can be derived from hPS cells using defined culture conditions (*Ng et al., 2008*; *Ng et al., 2005*) and can be identified by dual expression of the cell surface markers CD34 and CD45 (*Figure 8A*). We previously showed that a specific WNT signal involving WNT9A and FZD9 increased the efficiency of this differentiation protocol, as monitored by an increase in the yield of CD34/CD45 double positive cells (*Grainger et al., 2019*; *Richter et al.,*

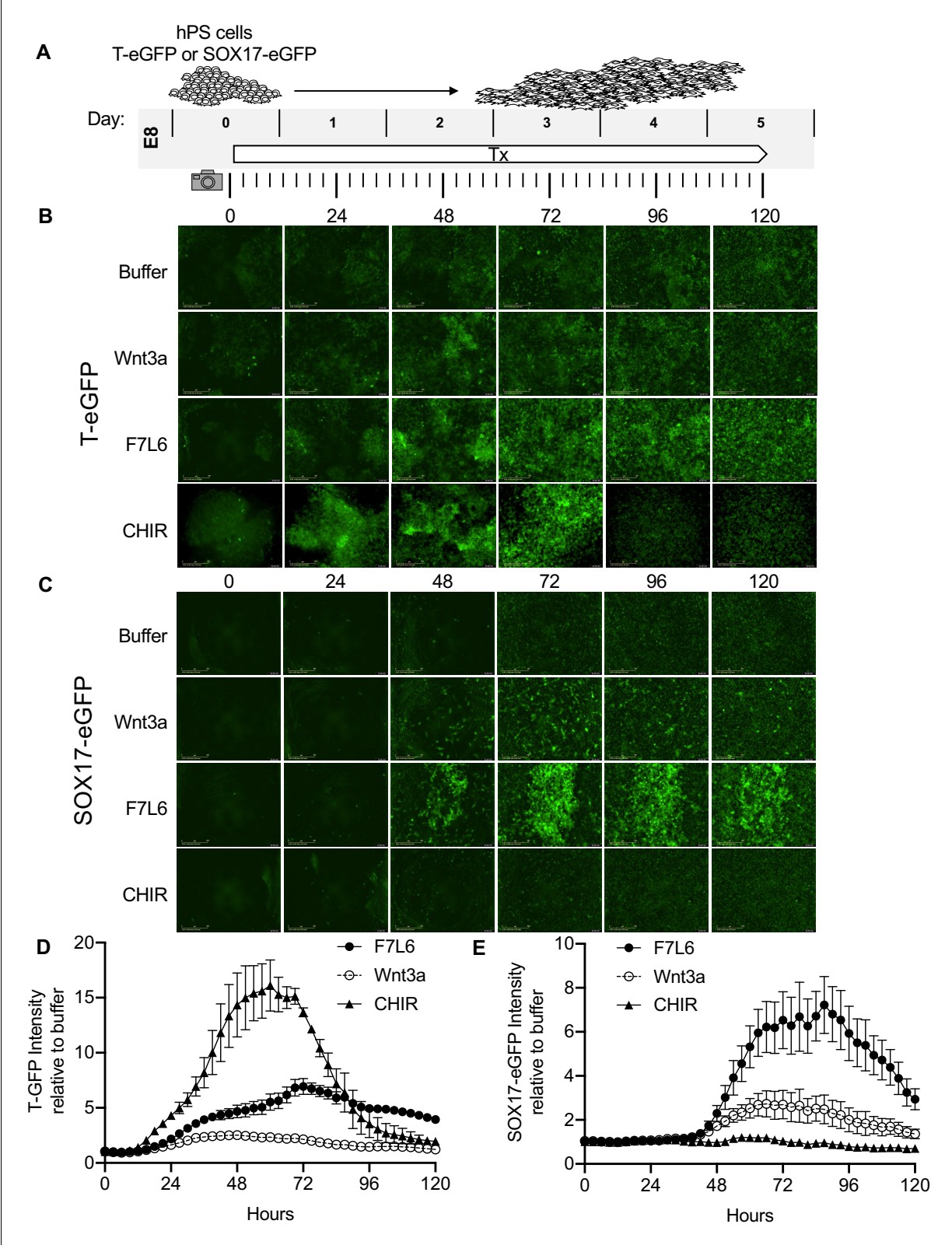

**Figure 6.** Activation of FZD7 with F7L6 promotes differentiation of hPS cells. (**A**) Schematic of live cell imaging experiment. Abbreviations: E8, essential eight medium; Tx, treatment. H9/WA09 cells carrying a T-GFP (**B, D**) or a SOX17-eGFP (**C, E**) reporter gene were treated with the indicated compounds, and fluorescence was imaged every 3 hr for a total of 120 hr on an IncuCyte Life Cell Analysis System. Fluorescence was quantified by total green object integrated intensity (GCU x μm$^2$/image).

*Figure 6 continued on next page*

*Figure 6 continued*

The online version of this article includes the following video(s) for figure 6:

**Figure 6—video 1.** Movies corresponding to *Figure 6B,D*.

https://elifesciences.org/articles/63060#fig6video1

**Figure 6—video 2.** Movies corresponding to *Figure 6C,E*.

https://elifesciences.org/articles/63060#fig6video2

*2018*). Given this highly selective requirement for WNT9A/FZD9 signaling, we reasoned that activation of FZD7 signaling with F7L6 would likely interfere with the differentiation of hPS cells toward the hematopoietic lineage. We found that F7L6 treatment at days 2–4 of differentiation, the treatment window we previously established to be critical for optimal differentiation, significantly reduced the number of CD34/CD45 double positive cells at day 14 relative to untreated or Wnt3a treated (*Figure 8B*). The percentages of CD34 and CD45 single positive cells were likewise adversely affected by F7L6 (*Figure 8C and D*). RT-qPCR demonstrated that F7L6 failed to induce expression of hematopoietic markers, including MESP1 and MIXL1 at day 4 and KDR at day 11 of differentiation (*Figure 8E*). Therefore, selective activation of FZD7 with F7L6 promotes differentiation into certain lineages, such as DE, but hinders differentiation into other lineages, such as the blood lineage.

## Discussion

The WNT/β-catenin signaling pathway is well known for its critical roles in stem cell biology, including maintenance of pluri- and multi-potency and differentiation. However, the study of individual ligand-receptor pairs in regulating stem cell behavior has been complicated by two main factors: first, the apparent promiscuity in WNT-receptor interactions and, second, the large number of *WNT* and *FZD* gene family members. Here, we used hPS cells to analyze the role of a single WNT receptor complex, comprised of FZD7 and LRP6, in regulating the transition from undifferentiated and pluripotent stem cell to a mesendodermally restricted cell lineage. Specific activation of signaling was achieved using a WNT mimetic, F7L6, which only engaged and thereby heterodimerized FZD7 and LRP6. In contrast to the widely used Wnt3a protein, which engages multiple cell surface receptors, F7L6 exclusively binds to FZD7 and LRP6.

We demonstrated that selective engagement and heterodimerization of FZD7 and LRP6 with F7L6 is sufficient to drive the differentiation program promoted by Wnt3a in hPS cells. FZD7 is the most abundantly expressed FZD receptor in hPS cells, and its knockdown by RNA interference disrupts expression of pluripotency-associated genes, such as *POU5F1*/*OCT4* and *NANOG* (*Fernandez et al., 2014*; *Melchior et al., 2008*). Likewise, inhibition of endogenous WNT processing in hPS cells using PORCN inhibitors also interferes with their pluripotency (*Fernandez et al., 2014*), suggesting that an endogenous WNT signal mediated by FZD7 is required to maintain hPS cells in an undifferentiated and pluripotent state. On the other hand, ectopic activation of WNT/β-catenin signaling, either with Wnt3a or with F7L6, leads to hPS cell differentiation along the mesendodermal lineage.

Whole transcriptome analysis revealed that Wnt3a and F7L6 elicit nearly identical transcriptional responses associated with early embryonic development, including formation of the primitive streak and gastrulation movements. Interestingly, we found that treatment with a GSK3 inhibitor (CHIR98014), which is commonly used interchangeably with Wnt3a, elicits significantly different effects. Most notably, we observed that over a 2-day treatment window, Wnt3a and F7L6 led to robust expression of the endodermal markers *FOXA2* and *SOX17*, whereas the GSK3 inhibitor did not. Consistent with these observations, Kafri et al. found that GSK3 inhibition elicited significantly different dynamics and kinetics of β-catenin accumulation and localization compared to Wnt3a (*Kafri et al., 2016*). These findings indicate that specific activation of WNT/FZD signaling is important to recapitulate developmental processes.

The WNT mimetic F7L6 has several advantages over recombinant WNT proteins. First, native WNT proteins interact promiscuously with multiple receptors. For example, Wnt3a interacts with FZD1, 2, 4, 5, 7, 8, 9, and 10 (*Dijksterhuis et al., 2015*; *Voloshanenko et al., 2017*). In addition, the co-receptors LRP5 and LRP6 are functionally redundant (*Kelly et al., 2004*), indicating that WNT proteins can interact equivalently with either one. In contrast, WNT mimetics can be engineered to

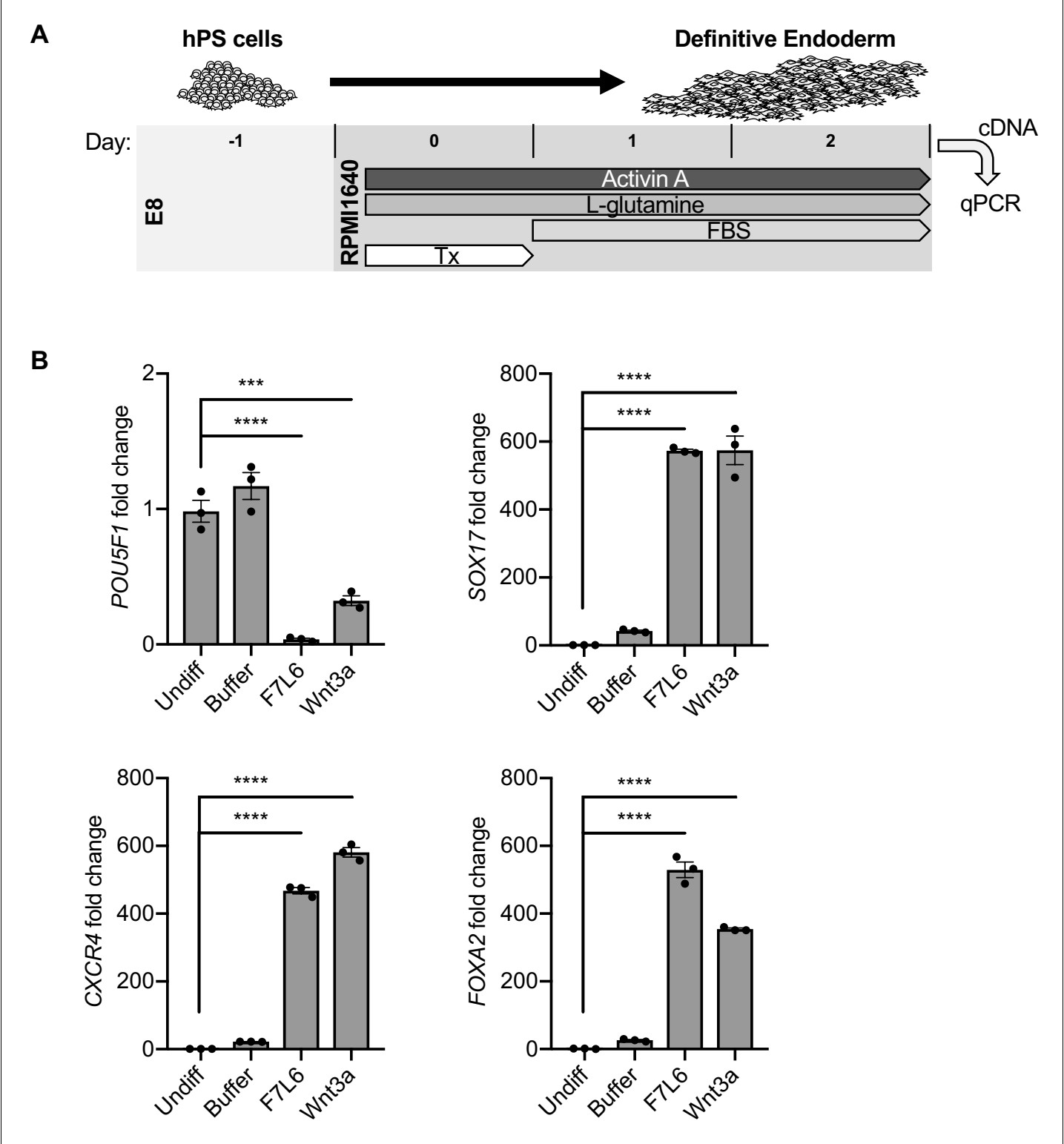

**Figure 7.** Activation of FZD7 with F7L6 promotes differentiation to definitive endoderm (DE). (**A**) Schematic of DE differentiation protocol. Abbreviations: E8, essential eight medium; FBS, fetal bovine serum; Tx, treatment. (**B**) RT-qPCR analysis of the differentiation treated with the indicated compounds. Treatment of hPS cells (H9/WA01) with F7L6 or Wnt3a leads to downregulation of the pluripotency marker *POU5F1* and upregulation of the DE markers *CXCR4*, *SOX17* and *FOXA2*. Gene expression was normalized to the expression of *RPL13A*. All samples were normalized to undifferentiated (Undiff) samples. For all statistical analyses: one-way ANOVA and Tukey's multiple comparisons test: ****p≤0.0001, ***p≤0.001. The online version of this article includes the following source data for figure 7:

**Source data 1.** Raw data for RT-qPCR assays shown in *Figure 7*.

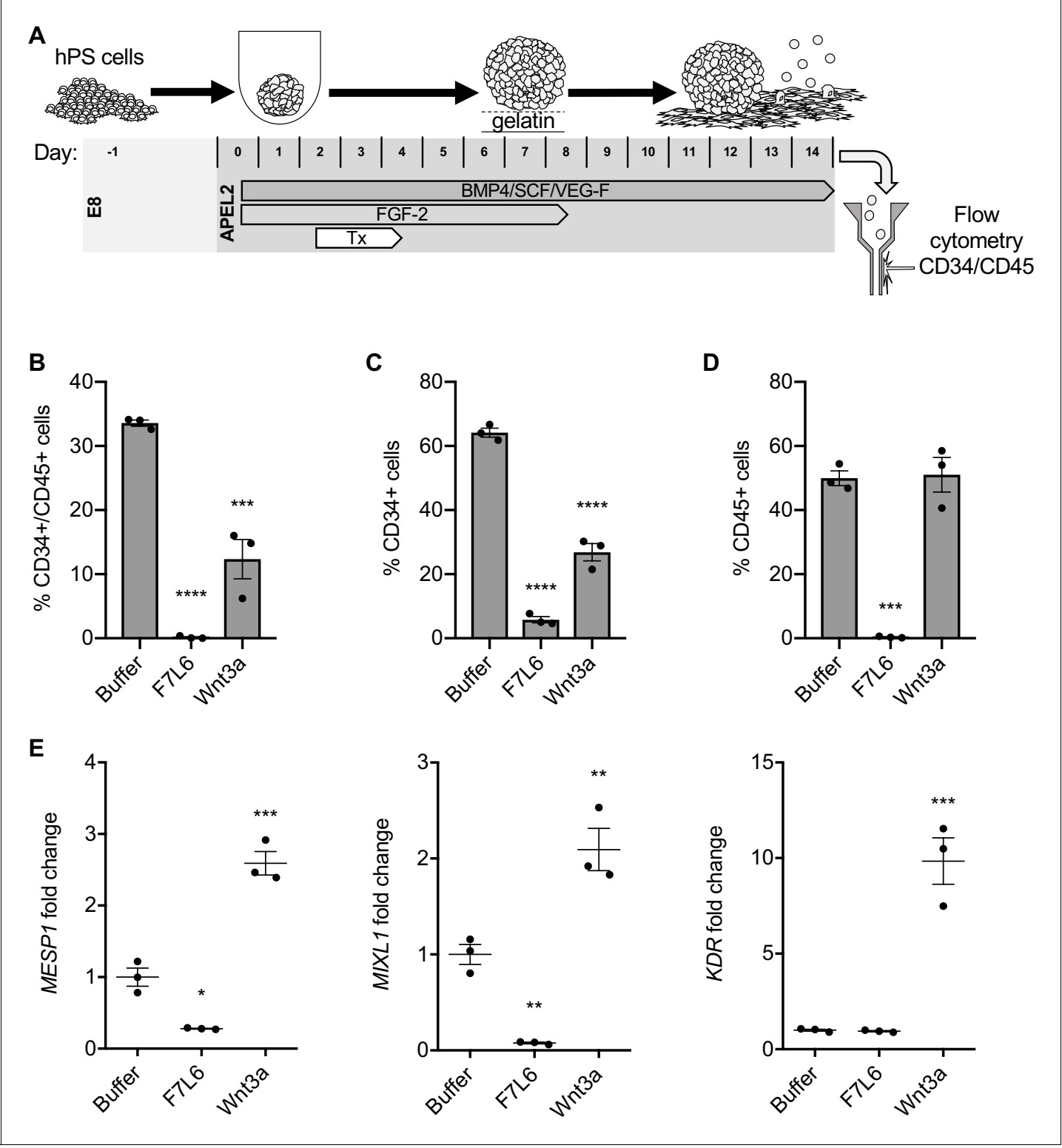

**Figure 8.** Activation of FZD7 with F7L6 blocks differentiation to hematopoietic stem and progenitor cells. (**A**) Schematic of the APEL hematopoietic stem/progenitor cell (HSPC) differentiation protocol. HPS cells were treated (Tx) with either Wnt3a or F7L6 from Days 2 to 4 of the 14 day differentiation protocol. On day 14, cells were analyzed by flow cytometry for the cell surface markers CD34 and CD45. Quantitation of flow cytometry of CD34/CD45 double positive cells (**B**), CD34 single positive cells (**C**), and CD45 single positive cells (**D**). (**E**) RT-qPCR analysis of MESP1 and MIXL1 at day 4 and of KDR at day 11 of differentiation. For statistical analyses: one-way ANOVA and Tukey's multiple comparisons test: ***p≤0.001, **p≤0.01, *p0.05, ns, not significant.

*Figure 8 continued on next page*

*Figure 8 continued*

The online version of this article includes the following source data for figure 8:

**Source data 1.** Raw data for flow cytometry and RT-qPCR assays shown in *Figure 8*.

interact with greater selectivity for cell surface receptors. In vivo, WNT-FZD signaling specificity is regulated in part by co-receptors. For example, Reck, a glycosylphosphatidylinositol-anchored cell surface protein, acts together with Gpr124, a 7-transmembrane protein, to promote Wnt7a-Fzd-Lrp5/6 signaling to promote angiogenesis in the developing central nervous system (*Cho et al., 2017*; *Cho et al., 2019*; *Eubelen et al., 2018*; *Vallon et al., 2018*). Furthermore, we recently identified a novel function for Egfr in mediating Wnt9a-Fzd9-Lrp5/6 signaling during hematopoietic development in zebrafish (*Grainger et al., 2019*). The use of engineered proteins capable of engaging specific WNT receptors, such as F7L6, may overcome the need for the recruitment of specificity-conferring co-receptors, such as Reck, Gpr124 and Egfr, and thus replace the need for purification of biologically active WNT proteins.

A second advantage of WNT mimetics is that they can be assembled using recombinant proteins with favorable biochemical properties. Purification of native WNT proteins has proven extremely difficult, requiring 3 to 4 chromatography steps with yields of approximately 0.1 mg per liter of WNT conditioned medium (*Willert, 2008*). Furthermore, WNT proteins are lipid modified (*Takada et al., 2006*; *Willert et al., 2003*), rendering them highly hydrophobic and necessitating the addition of detergents to maintain their solubility in aqueous conditions. Consistent with its poor physicochemical properties, at low concentrations the activity of Wnt3a is rapidly extinguished and nearly undetectable. In contrast, F7L6, which is entirely comprised of engineered immunoglobulin sequences and hence is soluble and stable in standard biologically compatible buffers, retains signaling activity at significantly lower concentrations. In addition, by appending tags, such as Fc or 6x-His, F7L6 can be purified using a single affinity binding step, with an approximate yield of 2.5 mg per liter of conditioned media.

Several other groups have described recombinant proteins similar in design to F7L6 and capable of heterodimerizing FZDs and LRP5/6 (*Chen et al., 2020*; *Dang et al., 2019*; *Janda et al., 2017*; *Miao et al., 2020*; *Tao et al., 2019*). In contrast to these other WNT agonists that engage the CRD of FZD, the FZD7-binding arm of F7L6 binds the linker, or neck region, between the CRD and the first transmembrane domain of FZD7. Since this neck region is poorly conserved among the ten FZD receptors, we were able to exclusively engage FZD7 and no other FZD, including mouse Fzd7. This feature of F7L6 allowed us to demonstrate that the CRD is dispensable for signaling, since F7L6, but not Wnt3a, could induce signaling by a CRD-less FZD7. Therefore, aside from docking a WNT protein, the CRD has no additional role in activating the downstream signaling events.

We designed F7L6 as a bispecific bivalent (=tetravalent) molecule capable of simultaneously binding two FZD7 and two LRP6 cell surface receptors. This design possibly permits WNT receptor oligomerization, which in turn promotes intracellular signalosome formation, as previously proposed (*DeBruine et al., 2017*). Structural studies have revealed that WNTs can dimerize FZD CRDs either in a 1:2 or a 2:2 stoichiometry (*Hirai et al., 2019*; *Nile et al., 2017*). These higher order oligomers are hypothesized to promote optimal downstream signaling. Interestingly, a bivalent version, F7L6-sc, capable of forming 1:1 FZD7-LRP6 heterodimers, promotes downstream signaling, indicating that higher order oligomerization is not strictly required for signaling, consistent with previous findings by others (*Miao et al., 2020*). Of note, signal saturation is approximately twofold greater for the tetravalent F7L6 version compared to the bivalent F7L6-sc version, indicating that oligomerization may control signal amplitude, possibly by polymerization of the intracellular scaffolding molecules DVL and AXIN to drive signalosome formation. Therefore, tetrameric ligands, such as F7L6 and those developed by others (*Chen et al., 2020*; *Tao et al., 2019*), may augment and potentiate signaling by promoting receptor clustering. Additional experiments are needed to resolve the contribution of WNT versus DVL/AXIN to receptor oligomerization and signalosome formation.

Despite their potent stem cell activities in vivo and their potential as therapeutics in regenerative medicine, WNT proteins have not yet been approved for any clinical applications, owing in part to their poor physicochemical properties. Certain formulations, such as liposomal packaging (*Morrell et al., 2008*), have produced more stable and bioactive WNT3A, which is in early-stage

clinical trials to treat patients undergoing posterolateral spinal fusion. However, such formulations will require purification of individual WNTs, which, to date, has only been achieved for WNT3A and WNT5A (*Mikels and Nusse, 2006*; *Willert et al., 2003*). WNT mimetics, such as the one described here, can be designed to target all WNT receptors and co-receptors and thus offers new opportunities for their development as therapeutics in regenerative medicine.

# Materials and methods

## Key resources table

| Reagent type (species) or resource | Designation | Source or reference | Identifiers | Additional information |
|---|---|---|---|---|
| Cell line (*Cricetulus griseus*) | CHO | ATCC | CCL-61 | RRID:CVCL_0213 |
| Cell line (*Homo sapiens*) | HEK293 | ATCC | CRL-1573 | RRID:CVCL_0045 |
| Cell line (*Homo sapiens*) | HEK293T-F127-KO | Prof. M. Boutros, Heidelberg University, Germany | | *Voloshanenko et al., 2017* |
| Cell line (*Homo sapiens*) | HEK293T-F124578-KO | Prof. M. Boutros, Heidelberg University, Germany | | *Voloshanenko et al., 2017* |
| Cell line (*Mus musculus*) | L1 | ATCC | CRL-2648 | RRID:CVCL_4536 |
| Cell line (*Homo sapiens*) | WA01 (H1) | WiCell Research Institute | NIH Registration Number: 0043 | Male, RRID:CVCL_9771 |
| Cell line (*Homo sapiens*) | WA09 (H9) | WiCell Research Institute | NIH Registration Number: 0062 | Female RRID:CVCL_9773 |
| Cell line (*Homo sapiens*) | H9 SOX17:GFP | Prof. S. Kim, Stanford University, USA | | *Wang et al., 2011* |
| Cell line (*Homo sapiens*) | H9 T-GFP | Prof. M. Mercola, Stanford University, USA | | *Kita-Matsuo et al., 2009* |
| Cell line (*Homo sapiens*) | iPS cells | Professor D. Kaufman, UCSD, USA | | *Li et al., 2018* |
| Strain, strain background (*Escherichia coli*) | BL21(DE3) | Invitrogen | C600003 | |
| Recombinant DNA reagent | Super TOPFlash (STF) | Addgene | Plasmid #12456 | RRID:Addgene_12456 |
| Recombinant DNA reagent | pFuse-hIgG1-Fc2 | Invivogen | pfuse-hg1fc2 | |
| Recombinant DNA reagent | pGEX-4T3 | Cytiva | 28-9545-52 | |
| Antibody | F7-Ab, chimeric human-mouse monoclonal | This paper | | 1 µg/mL, available upon request from corresponding author |
| Antibody | V5, mouse monoclonal | GeneTeX | Cat# GTX628529 | Immunoblot, 1:4000 Immunofluorescence, 1:500 |
| Antibody | Anti-β-catenin, mouse monoclonal | Sigma-Aldrich | Cat# C7207 | 1:2000 RRID:AB_476865 |
| Antibody | Anti-β-actin, mouse monoclonal | Sigma-Aldrich | Cat# A2228 | 1:5000 RRID:AB_476697 |
| Antibody | Goat anti-human IgG HRP-conjugated, goat polyclonal | ThermoFisher Scientific | Cat# 62–8420 | 1:20,000 RRID:AB_88136 |
| Antibody | Goat anti-mouse IgG HRP-conjugated, goat polyclonal | Southern Biotech | Cat#: 1030–05 | 1:20,000 RRID:AB_2619742 |
| Antibody | APC anti-human CD34, mouse monoclonal | Biolegend | Cat# 343608 | 1:100 RRID:AB_2228972 |

*Continued on next page*

*Continued*

| Reagent type (species) or resource | Designation | Source or reference | Identifiers | Additional information |
|---|---|---|---|---|
| Antibody | PE anti-human CD45, mouse monoclonal | Biolegend | Cat# 304008 | 1:100 RRID:AB_314396 |
| Antibody | APC mouse IgG2a k isotype control, mouse monoclonal | Biolegend | Cat# 400222 | 1:100 |
| Antibody | PE mouse IgG1, k isotype control, mouse monoclonal | Biolegend | Cat# 400112 | 1:100 RRID:AB_2847828 |
| Antibody | Alexa-Fluor 488 Goat anti-Mouse IgG, goat polyclonal | Invitrogen | Cat# A11001 | 1:1000 RRID:AB_2534069 |
| Recombinant protein | F7L6 | This paper | | Purified from CHO_His-F7L6-Fc, available upon request from corresponding author |
| Recombinant protein | F7L6-sc | This paper | | Purified from CHO_His-F7L6, available upon request from corresponding author |
| Recombinant protein | Wnt3a | Produced in Willert lab **Willert et al., 2003 Willert, 2008** | | Purified from CHO_Wnt3a cells |
| Recombinant protein | RSPO1 | Prof. Xi He, Harvard Medical School | | Purified from HEK293 _Rspo1 cells |
| Recombinant protein | FLAg $F^{P+P}$-$L6^{1+3}$ | Prof. S. Angers, Toronto University, Canada **Tao et al., 2019** | | |
| Recombinant protein | ActivinA | R and D Systems | Cat# 338-AC | 100 ng/mL |
| Recombinant protein | BMP4 | R and D Systems | Cat# 314 BP | 40 ng/mL |
| Recombinant protein | SCF | R and D Systems | Cat# 7466-SC | 40 ng/mL |
| Recombinant protein | VEGF | R and D Systems | Cat# 293-VE | 20 ng/mL |
| Recombinant protein | FGF2 | StemCell Technologies | Cat#78003 | 10 ng/mL |
| Commercial assay or kit | SuperSignal West Dura Western Blot Substrate | ThermoFisher Scientific | Cat# 34075 | |
| Commercial assay or kit | Pierce Coomassie (Bradford) Protein Assay Kit | ThermoFisher Scientific | Cat# 23200 | |
| Commercial assay or kit | TRIzol Reagent | ThermoFisher Scientific | Cat# 15596026 | |
| Commercial assay or kit | Direct-zol RNA MiniPrep Kit | Zymo Research | Cat# R2051 | |
| Commercial assay or kit | iScript Reverse Transcription Supermix | Bio-Rad | Cat# 1708840 | |
| Commercial assay or kit | iTaq Universal SYBR Green Supermix | Bio-Rad | Cat# 1725120 | |
| Chemical compound, drug | Zeocin | ThermoFisher Scientific | R25005 | 1 mg/mL |
| Chemical compound, drug | Puromycin | ThermoFisher Scientific | A1113802 | 4 µg/mL |
| Chemical compound, drug | Rock inhibitor Y-27631 | Tocris | Cat# 1254 | 5 µM |
| Chemical compound, drug | GSK3 inhibitor CHIR98014 | Sigma-Aldrich | Cat# SML1094 | 250 nM |
| Chemical compound, drug | ATP | Sigma-Aldrich | Cat# A2383 | |

*Continued on next page*

*Continued*

| Reagent type (species) or resource | Designation | Source or reference | Identifiers | Additional information |
|---|---|---|---|---|
| Chemical compound, drug | D-luciferin-Potassium Salt | ThermoFisher Scientific | Cat# 50227 | |
| Other | DAPI stain | Cell Signaling Technology | Cat# 4083S | (1 µg/mL) |
| Other | Protein G Sepharose | BioVision | Cat# 6511 | |
| Other | HiTrap IMAC HP, 1 mL | Cytiva | Cat# 17092003 | |
| Other | Superdex 200 10/300 GL | Cytiva | Cat# GE28-9909-44 | |
| Other | Matrigel | BD Biosciences | Cat# 356234 | |
| Other | mTeSR1 | StemCell Technologies | Cat# 85850 | |
| Other | APEL2 | StemCell Technologies | Cat# 05270 | |

## Design of F7, L6 and F7L6

The sequence coding for the heavy ($V_H$) and light ($V_L$) chain of the variably fragment (Fv) portion of the FZD7 antibody (F7-Ab) were identified and used to design a single-chain variable fragment (F7-Ab scFv) that consists of the heavy chain Fv fused to the light chain Fv with a linker peptide $(GGGGS)_3$. The F7-Ab scFv sequence was cloned into the pFuse-hIgG1-Fc2 mammalian expression vector (Invivogen), containing the IL-2 signal sequence and the IgG1 crystallizable fragment (Fc), to form the F7-Ab scFv-Fc (F7). The domains of the F7 construct are in the following order: IL-2 ss, F7-$V_H$, $(GGGGS)_3$, F7-$V_L$, Fc. The coding sequence for the LRP6 scFv (United States Patent No.: US8,883,735, SEQ ID NO: 81 $V_L$ and SEQ ID NO: 82 $V_H$) was similarly cloned into the pFuse mammalian expression vector to form the LRP6-Ab scFv-Fc (L6). The domains of the L6 construct are in the following order: IL-2 ss, L6-$V_L$, $(GGGGS)_4$, L6-$V_H$, Fc. The coding sequence for the two scFv were fused with an additional flexible linker and cloned into the pFuse mammalian expression vector to create a bispecific scFv-Fc (F7L6). The domains of the F7L6 construct are in the following order: IL-2 ss, F7-$V_H$, $(GGGGS)_3$, F7-$V_L$, $(GGGGS)_3$, L6-$V_L$, $(GGGGS)_4$, L6-$V_H$, Fc. Additional constructs containing a 6xHis tag between the IL-2 ss and the F7-$V_H$ domain with and without the Fc region were generated to express and produce F7L6 and F7L6-sc, respectively (*Supplementary file 4*).

## Expression, purification, and characterization of recombinant proteins

CHO cells (RRID:CVCL_0213) were transfected with plasmids encoding F7, L6, F7L6 and F7L6-sc, drug selected (1000 µg/mL zeocin or 4 µg/mL Puromycin, ThermoFisher Scientific) and expanded as clonal stable lines. Conditioned media (CM) were collected from confluent cultures every 3–4 days. Medium was replenished and CM was collected until the cells no longer adhered to the plate. CM were passed through a 0.22 µm filter (Genesee Scientific) and stored at 4°C until time of purification. CHO cells expressing the recombinant Wnt3a protein were similarly cultured and conditioned medium was collected, filtered, and stored. Wnt3a protein used in these studies was purified by four-step column chromatography as previously described (*Willert, 2008*). F7, L6, and F7L6 were purified from CM as follows: CM were applied to a protein G Sepharose (Biovision) column. The column was then extensively washed with phosphate buffered saline (PBS), and bound proteins were eluted with 0.1M glycine pH 2.5 and collected as 1 mL fractions into tubes containing 0.1 mL 2M Tris-Cl, pH8. Fractions containing the protein of interest were combined and dialyzed against PBS using dialysis tubing with a MWCO of 10 kDa (Thermo Scientific). After dialysis, the recombinant proteins were sterile filtered and aliquoted and stored at −80°C. The 6xHis-tagged version of F7L6 and F7L6-sc were purified as follows: CM from CHO cells expressing either F7L6 or F7L6-sc were adjusted to contain 30 mM imidazole and applied to a 1 mL immobilized metal affinity column charged with $NiCl_2$ (HiTrap IMAC HP, Cytiva Lifesciences). The column was washed with 30 mM imidazole in PBS. Bound proteins were eluted with a linear gradient from 30 mM to 300 mM imidazole in PBS into 1 mL fraction. Fractions containing F7L6 were combined and dialyzed against PBS. Fractions containing F7L6-sc were further fractionated by size exclusion chromatography (SEC, Superdex 200 10/300 GL, Cytiva Lifesciences) to remove contaminant proteins. After dialysis or gel filtration, the recombinant proteins were aliquoted and stored at −80°C. SEC of F7L6 and F7L6-sc

(as shown in *Figure 2—figure supplement 3*) was performed on a Superdex 200 10/300 GL column (Cytiva Lifesciences) in PBS, and 1 mL fractions were assayed in the presence of RSPO (100 ng/mL) for signaling activity using the STF assay. The previously published Wnt mimetic FLAg F$^{P+P}$-L6$^{1+3}$ (*Tao et al., 2019*) was generously provided by Stephane Angers (University of Toronto, Canada).

## Mapping of the F7-Ab epitope

The sequence encoding 42 amino acids (aa) of FZD7 containing the epitope for the F7-Ab (Glycine 168 to Serine 209) was cloned into the bacterial GST expression vector pGEX-4T3 (Cytiva Lifesciences). The epitope was shortened from either end through sequential cloning to produce 10 different length epitopes ranging from the original 42 aa to eight aa. The GST fusion proteins were expressed in BL21 cells and cell pellets were resuspended in protein sample loading buffer (2% SDS, 5% 2-mercaptoethanol, 10% glycerol, 62.5 mM Tris-Cl pH6.8) and boiled for 5 min (min) at 95°C. Proteins were visualized by Coomassie or by immunoblotting using the F7-Ab to determine binding to each epitope length.

## Cell lines and culture conditions

All cell lines used in these studies were directly obtained from either ATCC (CHO, HEK293, L) or WiCell Research Institute (H1, H9). Identity of H1 and H9 was authenticated by karyotyping and functional assays, including EB formation and directed differentiation. All HEK293/HEK293T (RRID: CVCL_0045/RRID:CVCL_0063) lines and mouse L-cells (ATCC-CRL-2648) (RRID:CVCL_4536) were cultured in Dulbecco's Modified Eagle's Medium (DMEM) supplemented with 10% fetal bovine serum (FBS) and penicillin/streptomycin. HEK293/HEK293T cells stably transduced with the Wnt reporter Super TOP-Flash (STF, Addgene Plasmid #12456, RRID:Addgene_12456) were previously described (*Bauer et al., 2013*). HEK293T cells carrying knockout mutations in LRP6 were previously described (*Grainger et al., 2019*). HEK293T harboring mutations in multiple FZD genes (FZD1,2,7 [F127-KO] and FZD1,2,4,5,7,8) were kindly provided by Professor M. Boutros, Heidelberg University, Germany (*Voloshanenko et al., 2017*). CHO cells overexpressing Wnt3a were cultured in DMEM, 10% FBS and penicillin/streptomycin, and Doxycycline (250 ng/mL) was added to induce Wnt3a expression.

All experiments using human pluripotent stem cell lines (hPS cells) were approved under IRB/ESCRO protocol number 100210 (Principal investigator: K.W.). HPS cell lines H1 (WA01, NIH Registration Number 0043, RRID:CVCL_9771) and H9 (WA09, NIH Registration Number 0062, RRID: CVCL_9773) were obtained from WiCell and cultured in E8 culture medium (*Chen et al., 2011*) on Matrigel (BD Biosciences). The H9 reporter lines, H9 SOX17:GFP (*Wang et al., 2011*) and H9 T-GFP (*Kita-Matsuo et al., 2009*) were cultured in E8 culture medium, passaged with TrypLE Express (Gibco), and seeded with Rock inhibitor Y-27631, 5 µM (Tocris). The H9 SOX17:GFP cell line was kindly provided by material transfer agreement by Dr. Seung Kim (Stanford School of Medicine). iPS cell lines were kindly provided by Dr. Dan Kaufman (UC San Diego) (*Li et al., 2018*). The iPS cells were cultured in mTeSR1 (StemCell Technologies), passaged with TrypLE Express, and seeded with Rock inhibitor. All hPS and iPS cells were fed fresh media daily.

## Differentiation of hPS cells

### Non-directed differentiation

Reporter hPS cell lines, H9 TBXT(T)-eGFP and H9 Sox17-eGFP, were passaged into a Matrigel-coated 12 well plate at 10,000 cells per well. When cells reached 40–60% confluence, media in each plate were replenished and the cells were treated with F7L6 (10 nM), Wnt3a (5 nM), CHIR98014 (250 nM, Sigma-Aldrich), or an equivalent volume of Wnt storage buffer (PBS, 1% CHAPS, 1 M NaCl). Plates were placed in the IncuCyte Life Cell Analysis System (Sartorius) and cultured without media changes for 5 days at 37°C. Phase and GFP images were recorded every 3 hr for a total of 120 hr after treatment. The IncuCyte software was used to process the images and generate the integrated GFP intensity (GCU x µm2/image).

### Endoderm differentiation

H1 cells were differentiated to endoderm as previously described (*D'Amour et al., 2006*). Initiated on days 4–6 after passage (depending on culture density), sequential, daily media changes were

made for the entire differentiation protocol. After a brief wash in PBS, cells were cultured in RPMI, Activin A (100 ng/mL, R and D Systems) and a treatment for the first day. Treatments were F7L6 (10 nM), Wnt3a (5 nM), CHIR (250 nM) or an equivalent volume of Wnt storage buffer. The next day the medium was changed to RPMI with 0.2% vol/vol FBS and Activin A (100 ng/mL), and the cells were cultured for 2 additional days. Cells were collected at day three and total RNA was isolated and analyzed by RT-qPCR.

## Hematopoietic differentiation

HPS cells were differentiated to the hematopoietic lineage as previously described (*Ng et al., 2008*; *Ng et al., 2005*). The day prior to differentiation, iPS cells were passaged with TrypLE Express at a high density in mTeSR with Rock inhibitor. On Day 0 of the differentiation, iPS cells were dissociated into a single cell suspension and plated in a 96-well U-bottom plate (Genesee Scientific) at 3000 cells per well in APEL2 (StemCell Technologies) with 40 ng/mL BMP4 (R and D Systems), 40 ng/mL SCF (R and D Systems), 20 ng/mL VEGF (R and D System), 10 ng/mL FGF-2 (StemCell Technology), and 5 nM Rock inhibitor. On Day 2 of the differentiation, the embryoid bodies (EB) were treated with F7L6 (5 nM), Wnt3a (5 nM), CHIR (250 nM) or an equivalent volume of Wnt storage buffer. On Day 4 of the differentiation, the media was removed from the EBs and replaced with fresh APEL2 with BMP4, SCF, VEGF, and FGF-2. On Day 7 of the differentiation, EBs were transferred to gelatin coated plates. On Day 8 of the differentiation one volume of APEL with BMP4, SCF, and VEGF was added. EBs were dissociated on Day 14 of the differentiation for analysis by flow cytometry.

## Immunoblotting by dot blot

Cells were lysed in TNT buffer (1% Triton X-100, 150 mM NaCl, 50 mM Tris HCl, pH 8) with protease inhibitors (Roche). Five µL of whole cell lysate (approximately 40 µg) was dotted directly onto nitrocellulose membrane and allowed to dry completely. The nitrocellulose membrane was incubated in blocking buffer (TBST [20 mM Tris-Cl pH8, 150 mM NaCl, 0.2% Tween-20], 1% BSA, 3% non-fat dry milk) for 30–60 min at room temperature (RT), and then incubated in primary antibody for 1 hr at RT or overnight at 4°C. The conditioned media collected from F7, L6, and F7L6 expressing CHO cells was mixed 1:1 with blocking buffer and used as a primary antibody solution. The nitrocellulose membrane was washed three times in TBST, and then incubated in goat anti-human HRP secondary antibody (ThermoFisher Scientific) at 1:10,000 dilution in blocking buffer for 45–60 min at RT. The nitrocellulose membrane was washed three times in TBST before enhanced chemiluminescent (ECL) detection by SuperSignal West Dura Western Blot Substrate (ThermoFisher Scientific) and exposure to autoradiography film.

## Immunoblotting by western

To obtain whole cell lysate, cells were lysed in TNT buffer with protease inhibitors. Protein concentrations were determined using Pierce Coomassie (Bradford) Protein Assay Kit (ThermoFisher Scientific). Protein sample loading buffer (62.5 mM Tris-Cl pH6.8, 2% SDS, 10% glycerol, 5% 2-mercaptoethanol, bromophenol blue) was added to 20 µg total protein. Samples were denatured at 95°C for 5 min unless samples were intended to be blotted for FZD proteins. Cell lysates were resolved by sodium dodecyl sulfate-polyacrylamide gel electrophoresis (SDS-PAGE), transferred to nitrocellulose membrane, and incubated for 30–60 min in blocking buffer. Primary antibody incubations were done overnight at 4°C. Primary antibodies and dilutions used: anti-FZD7 (F7-Ab) (1:1000 of 1 mg/mL stock), V5 tag antibody (GeneTex/GTX628529/1:4,000), monoclonal anti-β-catenin antibody (Sigma-Aldrich/C7207/1:2,000) (RRID_AB_476865), anti-β-actin antibody (Sigma-Aldrich/A2228/1:5,000) (RRID:AB_476697). All western blots were washed three times in TBST prior to incubation in secondary antibody for 45–60 min at RT. Secondary antibodies and dilutions used: goat anti-human IgG HRP-conjugated (ThermoFisher Scientific/62–8420/1:20,000) (RRID:AB_88136), goat anti-mouse IgG HRP-conjugated (Southern Biotech/1030–05/1:20,000) (RRID:AB_2619742). All western blots were washed three times in TBST before protein detection by Millipore Sigma Luminata Forte Western HRP substrate and exposure to autoradiography film.

## Super TOP-Flash (STF) luciferase assays

Cells were lysed in luciferase assay buffer (100 mM K-PO$_4$ buffer pH 7.8, 0.2% Triton X-100) and transferred to a black-walled 96-well plate. One hundred µL of luciferase assay cocktail was added to each well of lysate 25 mM Tris-Cl pH 7.8, 15 mM MgSO$_4$, 10 mM ATP (Sigma-Aldrich), 65 µM BD D-luciferin-Potassium Salt (Fisher Scientific). Luciferase assay readouts were performed on a Promega GloMax Discover Microplate Reader.

## Real-time quantitative polymerase chain reaction (RT-qPCR)

RNA expression was measured by RT-qPCR. RNA was extracted using TRIzol Reagent (ThermoFisher Scientific) and Direct-zol RNA MiniPrep Kit (Zymo Research). cDNA was generated using 50 ng RNA and iScript Reverse Transcription Supermix (Bio-Rad), then diluted 1:10 in UltraPure DNase/RNase-Free Distilled Water (ThermoFisher). RT-qPCR was performed using iTaq Universal SYBR Green Supermix (Bio-Rad) according to the manufacturer's recommendations, and a two-step amplification CFX_2stepAmp protocol on a Bio-Rad CFX384 Touch Real-Time PCR Detection System. Data was analyzed using the $2^{-\Delta\Delta Ct}$ method (*Schefe et al., 2006*). All gene expressions were normalized to the expression of *RPL13A*. The following RT-qPCR primers were used (Gene ID, forward primer, reverse primer):

ARL4A, GGCGATTTAGTCAAGAGGAT, GCTCTTCTCAACACACTACA
AXIN2, TATCCAGTGATGCGCTGACG, CGGTGGGTTCTCGGGAAATG
CXCR4, ACTACACCGAGGAAATGGGCT, CCCACAATGCCAGTTAAGAAGA
FGF18, GAGGAGAACGTGGACTTCCG, ACCTGGATGTGTTTCCCACT
FOXA2, GGAGCAGCTACTATGCAGAGC, CGTGTTCATGCCGTTCATCC
KDR, CCTGTATGGAGGAGGAGGAAGT, CAAATGTTTTTACACTCACAGGCCG
MESP1, CTCTGTTGGAGACCTGGATG, CCTGCTTGCCTCAAAGTG
MIXL1, TCCAGGATCCAGCTTTTATTTTCT, GAGGATAATCTCCGGCCTAGC
NKD1, TCCCAACCTAGAAACCTTAG, AGAAGAAGGAGAAGGAAGAG
NKX1-2, GTAGAAGAGAGGGAATAGGGAGAG, AGCAGCAGAAGTCCAAAGTC
POU5F1, CTTGAATCCCGAATGGAAAGGG, GTGTATATCCCAGGGTGATCCTC
RPL13A, CCTGGAGGAGAAGAGGAAAGAGA, TTGAGGACCTCTGTGTATTTGTCAA
SOX9, GACACAAACATGACCTATCC, GATTCTCCATCATCCTCCAC
SOX17, GTGGACCGCACGGAATTTG, GGAGATTCACACCGGAGTCA
SP5, TCGGACATAGGGACCCAGTT, CTGACGGTGGGAACGGTTTA

## Flow cytometry

HPS cell-derived EBs were dissociated with TrypLE Express at 37°C for 5–10 min, periodically triturated using P1000 pipette. Dissociated EBs were resuspended in FACS buffer (PBS, 1 mM EDTA, and 0.5% FBS) and passed through a cell strainer (Corning). Cell suspensions were pelleted at 200xg for 3 min and resuspended in 100 µL of FACS buffer (approximately 1 × 10$^5$ cells). Cells were incubated on ice for 30 min in the following primary fluorophore-conjugated antibodies (Antibody/Vendor/Catalog#/Concentration): APC anti-human CD34/Biolegend/343608/1:100 (RRID:AB_2228972); PE anti-human CD45/Biolegend/304008/1:100 (RRID:AB_314396); APC mouse IgG2a k isotype control/Biolegend/400222/1:100; PE mouse IgG1, k isotype control/Biolegend/400112/1:100 (RRID:AB_2847828). Cells were washed with 2 mL of FACS buffer and spun down at 200xg for 3 min. Cells were resuspended in 500 mL of FACS buffer with 0.5 µg/mL DAPI (Cell Signaling Technology). The cell suspensions were analyzed using the FACS Fortessa (BD Biosciences) and the resulting FSC files were processed with the FlowJo software (BD Biosciences).

## Immunofluorescence of overexpressed FZD

HEK293T F127-KO cells were plated in a 24 well-plate with coverslips. Cells were transfected with V5-tagged FZD receptors and 2 days post-transfection, cells were fixed with 4% formaldehyde for 15 min at RT. Cells were washed twice with 1X PBS and permeabilized with 1X PBS with 0.5% Triton X-100 for one hour at RT. Following permeabilization, cells were blocked with 0.5% BSA in 1X PBS with 0.1% Triton X-100 (1X PBST) for 1 hr at RT. Cells were stained with V5 antibody (Genetex/GTX628529/1:500) in 1X PBST at 4°C overnight. Cells were washed three times with 1X PBST for 10 min each. Alexa-Fluor 488 Goat anti-Mouse IgG (Invitrogen/A11001/1:1000) (RRID:AB_2534069) in

1X PBST was used as the secondary antibody. Secondary antibody staining was done for 2 hr at RT. Following the secondary staining, cells were washed three times with 1X PBST for 10 min each. Coverslips were mounted on glass slides and imaged on Nikon Eclipse Ti2-E microscope with SR HP Apo TIRF 100x NA 1.49 objective.

### Transcriptome analysis (RNA-seq)

Total RNA from cells was extracted using TRIzol Reagent and Direct-zol RNA MiniPrep Kit, according to manufacturer recommendations. cDNA library preparation and sequencing were done by Novogene Co., Ltd. A 250–300 bp insert cDNA library was generated by using the NEBNext Ultra RNA Library Prep Kit for Illumina (New England Biolabs, Inc). Transcriptome sequencing was performed on an Illumina NovaSeq 6000. TopHat (RRID:SCR_013035) and Cufflinks (RRID:SCR_014597) (*Trapnell et al., 2013*; *Trapnell et al., 2012*) were used to perform differential gene expression analysis of RNA-seq experiments. Briefly, sequencing reads were quality filtered, mapped, and aligned to the reference human genome (hg19) with TopHat and Cuffdiff (RRID:SCR_001647) was used to calculate gene expression levels as reads per thousand transcript bases per million reads mapped (RPKM). Statistically significant changes in gene expression were obtained from RPKM values. Genes were clustered by expression pattern and principal component analysis was performed in Genesis (*Sturn et al., 2002*) (RRID:SCR_015775). Gene ontology was performed using GSEA (*Mootha et al., 2003*; *Subramanian et al., 2005*) (RRID:SCR_003199).

## Acknowledgements

Research reported in this publication was supported by the National Institute of General Medical Sciences of the National Institutes of Health under Award Number R35GM134961 (awarded to KW). The content is solely the responsibility of the authors and does not necessarily represent the official views of the National Institutes of Health. MD was supported by in part by a Cancer Biology, Informatics, Omics Training Program from the National Cancer Institute (T32 CA067754). This work was additionally made possible by the CIRM Major Facilities grant (FA1-00607) to the Sanford Consortium for Regenerative Medicine. This publication includes data generated at the UCSD Human Embryonic Stem Cell Core Facility utilizing the IncuCyte Zoom Live Cell Analysis System that was purchased with funding from a NIH Shared Instrumentation Grant (S10OD025060), and at the UC San Diego IGM Genomics Center utilizing an Illumina NovaSeq 6000 that was purchased with funding from a NIH Shared Instrumentation Grant (S10OD026929).

## Additional information

### Funding

| Funder | Grant reference number | Author |
| --- | --- | --- |
| National Institutes of Health | R35GM134961 | Karl Willert |
| National Institutes of Health | S10OD026929 | Karl Willert |
| National Cancer Institute | T32 CA067754 graduate student fellowship to Myan Do | Myan Do |
| NIH | S10OD025060 | Karl Willert |
| National Institutes of Health | R01HL135205 | Karl Willert |

The funders had no role in study design, data collection and interpretation, or the decision to submit the work for publication.

### Author contributions

Diana Gumber, Data curation, Investigation, Methodology, Writing - original draft, Writing - review and editing; Myan Do, Data curation, Investigation, Methodology, Writing - original draft; Neya Suresh Kumar, Pooja R Sonavane, Christina C N Wu, Luisjesus S Cruz, Investigation, Methodology; Stephanie Grainger, Supervision, Investigation, Methodology; Dennis Carson, Conceptualization,

Supervision, Writing - review and editing; Terry Gaasterland, Data curation, Software, Formal analysis, Supervision; Karl Willert, Conceptualization, Supervision, Funding acquisition, Writing - original draft, Project administration, Writing - review and editing

## Author ORCIDs
Diana Gumber 🆔 https://orcid.org/0000-0002-0913-8001
Myan Do 🆔 http://orcid.org/0000-0001-5892-6859
Karl Willert 🆔 https://orcid.org/0000-0002-8020-6804

## Decision letter and Author response
Decision letter https://doi.org/10.7554/eLife.63060.sa1
Author response https://doi.org/10.7554/eLife.63060.sa2

---

# Additional files

## Supplementary files
• Supplementary file 1. RNA-seq data set for all significantly differentially expressed genes in hPS cells (H1/WA01) treated with F7L6, Wnt3a, or CHIR. (A) Reads per kilobase per million mapped reads (RPKM) values for 1814 genes with significant fold changes in expression in response to F7L6, Wnt3a, or CHIR. (B) 525 genes with significant fold changes in expression in response to F7L6. (C) 708 genes with significant fold changes in expression in response to Wnt3a. (D) 1646 genes with significant fold changes in expression in response to CHIR. (E) 428 genes overlapping in F7L6 and Wnt3a treatments, with significant fold changes in expression. (F) RPKM values for 805 genes with significant fold changes in expression in response to F7L6 or Wnt3a. (G) 1009 genes unique to CHIR treatment, with significant fold changes in expression. (H) 391 genes overlapping across F7L6, Wnt3a, and CHIR treatments, with significant fold changes in expression. (I) 411 genes overlapping in F7L6 and CHIR treatments, with significant fold changes in expression. (J) 617 genes overlapping in Wnt3a and CHIR treatments, with significant fold changes in expression. (K) 77 genes unique to F7L6 treatment with significant fold changes in expression. (L) 54 genes unique to Wnt3a treatment, with significant fold changes in expression.

• Supplementary file 2. Gene Set Enrichment Analysis (GSEA) data for genes significantly differentially expressed in response to F7L6, Wnt3a, and/or CHIR in hPS cells (H1/WA01). (A) Top ten gene set hits from GSEA for overlapping genes significantly differentially expressed in response to F7L6, Wnt3a, and CHIR. The corresponding gene list is provided in *Supplementary file 1H*. (B) Top ten gene set hits from GSEA for genes significantly differentially expressed in response to F7L6. The corresponding gene list is provided in *Supplementary file 1B*. (C) Top ten gene set hits from GSEA for genes significantly differentially expressed in response to Wnt3a. The corresponding gene list is provided in *Supplementary file 1C*. (D) Top 10 gene set hits from GSEA for genes significantly differentially expressed in response to CHIR. The corresponding gene list is provided in *Supplementary file 1D*. (E) Top 10 gene set hits from GSEA for unique genes significantly differentially expressed in response to CHIR. The corresponding gene list is provided in *Supplementary file 1G*. (F) Top 10 gene set hits from GSEA for overlapping genes significantly differentially expressed in response to F7L6 and Wnt3a. The corresponding gene list is provided in *Supplementary file 1E*. (G) Top 10 gene set hits from GSEA for overlapping genes significantly differentially expressed in response to F7L6 and CHIR. The corresponding gene list is provided in *Supplementary file 1I*. (H) Top 10n gene set hits from GSEA for overlapping genes significantly differentially expressed in response to Wnt3a and CHIR. The corresponding gene list is provided in *Supplementary file 1J*.

• Supplementary file 3. Lists of genes activated by Wnt3a in H1/WA01 at 3 hr (A) and 6 hr (B). Expression level changes for these gene sets are shown in *Figure 5A and B*.

• Supplementary file 4. Lists of plasmids available upon request from corresponding author.

• Transparent reporting form

## Data availability

The RNA-seq and ChIP-seq data discussed in this publication have been deposited in NCBI's Gene Expression Omnibus and are accessible through GEO Series accession number GSE158121.

The following dataset was generated:

| Author(s) | Year | Dataset title | Dataset URL | Database and Identifier |
|---|---|---|---|---|
| Gaasterland T, Willert K | 2020 | Selective activation of FZD7 in human pluripotent stem cells | https://www.ncbi.nlm. nih.gov/geo/query/acc. cgi?acc=GSE158121 | NCBI Gene Expression Omnibus, GSE158121 |

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
