## [Decision Letter]

**Acceptance summary:**

This paper provides convincing evidence that F7L6 is a highly useful synthetic ligand for the dissection of Wnt signaling in vivo and in vitro, adding important data to the growing literature on synthetic Wnt ligands. This work will be of interest to those working on Wnt signaling and especially for those who are interested in harnessing Wnt signaling reagents for practical purposes.

**Decision letter after peer review:**

Thank you for submitting your article "Selective activation of FZD7 promotes mesendodermal differentiation of human pluripotent stem cells." for consideration by *eLife*. Your article has been reviewed by three peer reviewers, including Yi Arial Zeng as the Reviewing Editor and Reviewer #1, and the evaluation has been overseen by Edward Morrisey as the Senior Editor.

The reviewers have discussed the reviews with one another and the Reviewing Editor has drafted this decision to help you prepare a revised submission.

Gumber et al., generated a WNT agonist, F7L6, that selectively engages FZD7 and co-receptor LRP6. They provided evidence for the specificity on Fzd7, and efficacy in turning on Wnt signaling. Then, they compared F7L6 with Wnt3a and CHIR, and exploited the utility of F7L6 in differentiation of hPSC. They reached conclusions that treatment of F7L6 directs endodermal differentiation of hPSC to a similar extend to as Wnt3a. The specificity and significance of F7L6 were further exemplified using the hematopoietic lineage differentiation of hPSC, a procedure selectively required for WNT9A/FZD9 signaling. They showed that F7L6 hinders differentiation into the blood lineage. Overall, the study is well organized and clearly presented. This manuscript adds important data to the growing literature on synthetic Wnt ligands, showing that this is a generalizable strategy and that F7L6 is a highly useful synthetic ligand for the dissection of Wnt signaling in vivo and in vitro. This work will be of interest to those working on Wnt signaling and especially for those who are interested in harnessing Wnt signaling reagents for practical purposes.

Essential revisions:

1) The results of the experiment in Figure 8 look odd because the buffer alone and Wnt3a groups each have 3 outliers. Without those outliers, the statistical difference compared with F7L6 treatment might be substantially different. On the one hand, one can say that the data is what it is and leave it at that, but I find that approach unsatisfying if the data suggests (as it does here) that there is some underlying variable which is not being well controlled. My suggestion would be to redo the experiment with many replicates and try to control all of the variables. In addition, it is a bit thin on the characterization of mesodermal/blood differentiation. I suggest to evaluate more markers, e.g. Mixl1, Mes1, Pdgfra or KDR.

2) To determine the specificity of Fzd7-Ab, the authors perform the following:

"Figure 1G. F7-Ab is specific to human. FZD7 and does not cross-react with the other nine FZDs (1-6, 8-10). F127-KO were transfected with expression constructs carrying the indicated human FZD cDNAs tagged with an intracellular V5 sequence. Non-permeabilized cells were stained with F7-Ab for cell-surface FZD expression, and then permeabilized and stained for V5 expression. All FZD receptors were expressed as

revealed by anti-V5 antibody staining."

– Strictly speaking, the presence of V5 tag doesn't indicate the presence of receptor on the membrane. Since this is the major/only assay determining Fzd specificity, can the author confirm that all Fzd receptors are indeed expressed on the cell surface, e.g., by FACS or V5 surface staining.

3) The authors stated that heterodimerization of LRP6 and Fzd7 is sufficient for pathway activation and also illustrated a Lrp6-Fzd (1:1) interaction on Figure 2A, while other studies claimed that tetrameric ligands are required. Notably, the construct used in this study is a Fc-dimer.

– Can the authors clarify if the F7L6 his-tag construct (without the Fc) is able to activate Wnt pathway robustly, i.e., the F7L6 construct is optimally designed to engage Fzd-neck region and LRP6 domain. Please indicate which LRP6 domain is targeted by the antibody.

4) This is a very valuable resource for the community and in light of previous publications, it would be helpful for potential users, but not required for the current paper, if the authors can compare the potency (EC50) of F7L6 with previously published reagents such as the NGS-Wnt (Miao et al., CSC, 2020), FLAg (Tao et al., 2019), and Wnt3a, for instance with HEK293-STF dose response curves (with and without Rspo).

5) RNA seq data comparing F6L7, Wnt3a and CHIR:

"In contrast to Wnt3a and F7L6, CHIR treatment produced significantly different gene expression profiles, both in kinetics and amplitude (Figure 3A, B)… Such differences are likely attributable to the fact that CHIR acts downstream of the receptor complexes and hence is not restricted by receptor availability."

– If the receptor availability accounts for the lower amplitude in responses when using Wnt/F6L7, as the author suggested, can this be addressed by using higher concentrations of ligands (Wnt/F7L6) or by adding R-spondin? Is Rspo activity on hPSC known? LGR4 transcripts appeared to be detected on the RNA-seq data.

– In addition, Kafri et al., previously showed that GSK3 inhibitors (in this case LiCl) can cause a more prolonged nuclear and cytoplasmic β-catenin accumulation (10 hours) than Wnt3a (2 – 3 hours) (https://elifesciences.org/articles/16748). Any thoughts on how this may affect the gene expression profile?

---

## [Author Response]

Essential revisions:1) The results of the experiment in Figure 8 look odd because the buffer alone and Wnt3a groups each have 3 outliers. Without those outliers, the statistical difference compared with F7L6 treatment might be substantially different. On the one hand, one can say that the data is what it is and leave it at that, but I find that approach unsatisfying if the data suggests (as it does here) that there is some underlying variable which is not being well controlled. My suggestion would be to redo the experiment with many replicates and try to control all of the variables. In addition, it is a bit thin on the characterization of mesodermal/blood differentiation. I suggest to evaluate more markers, e.g. Mixl1, Mes1, Pdgfra or KDR.

We agree with the reviewer that the data presented in Figure 8 was quite noisy. We have since redone this experiment with additional replicates and thereby managed to reduce the high degree of variability observed in our earlier attempts. This new data is presented as new Figure 8B-D. In addition, we provide further characterization of the differentiation using RT-qPCR analysis of hematopoietic markers MESP1 and MIXL1 at day 4 and KDR at day 11 of differentiation. These data demonstrate that treatment with F7L6 of hPS cells disrupts their ability to differentiate into the blood lineage.

2) To determine the specificity of Fzd7-Ab, the authors perform the following:"Figure 1G. F7-Ab is specific to human. FZD7 and does not cross-react with the other nine FZDs (1-6, 8-10). F127-KO were transfected with expression constructs carrying the indicated human FZD cDNAs tagged with an intracellular V5 sequence. Non-permeabilized cells were stained with F7-Ab for cell-surface FZD expression, and then permeabilized and stained for V5 expression. All FZD receptors were expressed asrevealed by anti-V5 antibody staining."– Strictly speaking, the presence of V5 tag doesn't indicate the presence of receptor on the membrane. Since this is the major/only assay determining Fzd specificity, can the author confirm that all Fzd receptors are indeed expressed on the cell surface, e.g., by FACS or V5 surface staining.

We agree with the reviewer that our flow cytometry data in Figure 1G does not establish that all FZD receptors are expressed on the cell surface and consequently would not be accessible to the FZD7 antibody (F7-Ab). To address this point, we provide additional confocal microscopy data in Figure 1—figure supplement 2. This analysis demonstrates that the V5-tagged FZD receptors are at the cell surface and hence are directly exposed to F7-Ab. We contend that these data unequivocally establish F7-Ab’s specificity for FZD7.

3) The authors stated that heterodimerization of LRP6 and Fzd7 is sufficient for pathway activation and also illustrated a Lrp6-Fzd (1:1) interaction on Figure 2A, while other studies claimed that tetrameric ligands are required. Notably, the construct used in this study is a Fc-dimer.– Can the authors clarify if the F7L6 his-tag construct (without the Fc) is able to activate Wnt pathway robustly, i.e., the F7L6 construct is optimally designed to engage Fzd-neck region and LRP6 domain. Please indicate which LRP6 domain is targeted by the antibody.

We appreciate the importance of this issue and have added additional experiments shown in Figure 2I and Figure 2—figure supplement 3. Due to the addition of an Fc portion, our Wnt mimetic F7L6 is bispecific and bivalent (i.e. tetravalent) and thus is potentially capable of simultaneously engaging two FZD7 and two LRP6 molecules. We have now generated a new WNT mimetic lacking the Fc portion such that the resulting protein is a bispecific binder that engages only one FZD7 and one LRP6 molecule. This bispecific protein, referred to as F7L6-sc (single chain), is biologically active, albeit slightly less active than the original F7L6 molecule. Furthermore, using size exclusion chromatography, we demonstrate that F7L6-sc is monomeric and does not homodimerize in solution. These data indicate that higher order receptor clustering is not strictly required for signal activation and possibly controls signal amplitude.

4) This is a very valuable resource for the community and in light of previous publications, it would be helpful for potential users, but not required for the current paper, if the authors can compare the potency (EC50) of F7L6 with previously published reagents such as the NGS-Wnt (Miao et al., CSC, 2020), FLAg (Tao et al., 2019), and Wnt3a, for instance with HEK293-STF dose response curves (with and without Rspo).

We have performed additional experiments to address these points. First, we obtained a previously published Wnt mimetic, the pan-specific FLAg F^P+P^-L6^1+3^ (Tao et al., 2019). In a side-by-side comparison, both Wnt mimetics are potent activators of the Super-TOP-Flash (STF) reporter with equivalent EC50s in the single digit nano-molar range. Second, we compared the signaling activities of F7L6 and Wnt3a. Both proteins activate the STF reporter, however, at low concentrations F7L6 significantly outperforms Wnt3a. This difference is likely attributable to Wnt3a’s poor chemicophysical properties, such as its hydrophobicity. Third, we performed all STF assays with and without RSPO1. This analysis confirms that RSPO potently augments the activity of Wnt3a, F7L6 and FLAg F^P+P^-L^1+3^. Of note, addition of RSPO1 increased F7L6 activity approximately 10-fold, but only augmented the activities of Wnt3a or FLAg F^P+P^-L^1+3^ by 2-fold. A possible reason for this distinction is that F7L6 is selective for FZD7, whereas Wnt3a and FLAg F^P+P^-L^1+3^ are capable of interacting with multiple FZDs. These new data are displayed in Figure 2C and Figure 2—figure supplement 1.

5) RNA seq data comparing F6L7, Wnt3a and CHIR:"In contrast to Wnt3a and F7L6, CHIR treatment produced significantly different gene expression profiles, both in kinetics and amplitude (Figure 3A, B)… Such differences are likely attributable to the fact that CHIR acts downstream of the receptor complexes and hence is not restricted by receptor availability."– If the receptor availability accounts for the lower amplitude in responses when using Wnt/F6L7, as the author suggested, can this be addressed by using higher concentrations of ligands (Wnt/F7L6) or by adding R-spondin? Is Rspo activity on hPSC known? LGR4 transcripts appeared to be detected on the RNA-seq data.

Based on dose response curves using SP5 expression as a readout (Figure 2H), we treated hPS cells with saturating concentrations of Wnt3a and F7L6 (each at 5nM) such that higher levels of ligands are unlikely to alter the response. However, we acknowledge the possibility that other target genes may require higher doses of these ligands. Performing additional transcriptome-wide expression analyses with higher ligand concentrations in the presence or absence of Rspo, while certainly of great interest, will require substantial resources and further extend the timeframe for revision. We therefore respectfully ask that such experiments are not required.

– In addition, Kafri et al., previously showed that GSK3 inhibitors (in this case LiCl) can cause a more prolonged nuclear and cytoplasmic β-catenin accumulation (10 hours) than Wnt3a (2 – 3 hours) (https://elifesciences.org/articles/16748). Any thoughts on how this may affect the gene expression profile?

This is an interesting point that deserves further investigation. In our current study, we focused on the transcriptional response to pathway activation rather than on β-catenin accumulation, and therefore, we can’t draw any conclusions about the timing of β-catenin accumulation and localization. We can only speculate that prolonged activation of the pathway will promote secondary and tertiary transcriptional events that may not involve the transcriptional activity of β-catenin. We agree that findings by Kafri et al., are potentially relevant to our studies and may explain the observed differences in transcriptional effects by GSK3 inhibitors versus Wnt3a or F7L6. Therefore, we have included a reference to this study in the Discussion section.